# A Closer Look at Graph Transformers: Cross-Aggregation and Beyond

**Jiaming Zhuo[1], Ziyi Ma[1], Yintong Lu[1], Yuwei Liu[1], Kun Fu[1], Di Jin[2],**
**Chuan Wang[3], Wenning Wu[4], Zhen Wang[4], Xiaochun Cao[5], Liang Yang[1]***
[1]Hebei Province Key Laboratory of Big Data Calculation,
School of Artificial Intelligence, Hebei University of Technology, Tianjin, China
[2]College of Intelligence and Computing, Tianjin University, Tianjin, China
[3]School of Computer Science and Technology, Beijing JiaoTong University, Beijing, China
[4]School of Artificial Intelligence, OPtics and ElectroNics (iOPEN),
School of Cybersecurity, Northwestern Polytechnical University, Xi'an, China
[5]School of Cyber Science and Technology,
Shenzhen Campus of Sun Yat-sen University, Shenzhen, China
`jiaming.zhuo@outlook.com, zyma@hebut.edu.cn,`
`{202332803037, 202322802030}@stu.hebut.edu.cn, fukun@hebut.edu.cn,`
`jindi@tju.edu.cn, wangchuan@iie.ac.cn, wuwenning@nwpu.edu.cn,`
`w-zhen@nwpu.edu.cn, caoxiaochun@mail.sysu.edu.cn, yangliang@vip.qq.com`

## Abstract

Graph Transformers (GTs), which effectively capture long-range dependencies and structural biases simultaneously, have recently emerged as promising alternatives to traditional Graph Neural Networks (GNNs). Advanced approaches for GTs to leverage topology information involve integrating GNN modules or modulating node attributes using positional encodings. Unfortunately, the underlying mechanism driving their effectiveness remains insufficiently understood. In this paper, we revisit these strategies and uncover a shared underlying mechanism—Cross Aggregation—that effectively captures the interaction between graph topology and node attributes. Building on this insight, we propose the Universal Graph Cross-attention Transformer (UGCFormer), a universal GT framework with linear computational complexity. The idea is to interactively learn the representations of graph topology and node attributes through a linearized Dual Cross-attention (DCA) module. In theory, this module can adaptively capture interactions between these two types of graph information, thereby achieving effective aggregation. To alleviate overfitting arising from the dual-channel design, we introduce a consistency constraint that enforces representational alignment. Extensive evaluations on multiple benchmark datasets demonstrate the effectiveness and efficiency of UGCFormer.

## 1 Introduction

Node classification, aimed at accurately predicting node categories based on the graph topology and node attributes, is a fundamental task in identifying the properties of individual nodes [12, 18, 14, 32, 11, 10]. As a powerful class of models for fusing topology and attribute information in graphs, Graph Neural Networks (GNNs) have achieved initial successes in this task [29, 5, 52, 24, 22, 25]. In general, they follow the graph-bound Message Passing (MP) paradigm [16]. While this paradigm endows GNNs with the localizing property, it also restricts their ability to capture long-range dependencies [9], resulting in well-known challenges such as over-smoothing [6, 59] and over-squashing [17].

---

*Corresponding author

39th Conference on Neural Information Processing Systems (NeurIPS 2025).

Inspired by the remarkable success of Transformers in NLP [35], Graph Transformers (GTs) have emerged as powerful architectures for node classification tasks. The core component of Transformers is the Self-Attention (SA) module [46], which models full interactions among tokens within a sequence, thereby endowing the Transformers with globalizing properties. The initial success of GTs can be attributed to the strategic integration of discriminative graph topology into Transformer architectures, enabling the simultaneous capture of structural biases and long-range dependencies. To date, two primary strategies have achieved SOTA performance in existing GTs: (1) integrating GNN blocks [31, 50, 7, 61], and (2) modulating node attributes utilizing Positional Encodings (PEs) [2, 49, 44]. However, both strategies face inherent limitations. The first tends to inherit drawbacks from GNNs due to its reliance on them, whereas the second introduces additional computational complexity due to the use of PEs, thereby restricting the models' universality[1] and scalability.

This leads to a fundamental question:

> *What underlying mechanism drives the effectiveness of diverse Graph Transformers?*

A thorough understanding of the underlying mechanisms can offer valuable insights for developing more advanced and efficient architectures. Following this line, this paper theoretically investigates the mechanism shared by the aforementioned types of GTs and, based on this insight, proposes a novel GT architecture. In particular, the unified cross-aggregation mechanism (as formally defined in Definition 1) is explored by analytically decoupling topology and attribute representations from node representations. Specifically, the GNN block in GTs can be interpreted as aggregating topology representations into attribute representations (in Theorem 1), indicating that this category of GTs inherently incorporates cross-aggregation. Furthermore, GTs employing PEs contain diverse forms of cross-aggregation between topology and attribute representations. Therefore, the shared underlying mechanism among these GTs is cross-aggregation between graph topology and node attributes.

This understanding naturally leads to a key question:

> *How can we design an effective and efficient GT architecture grounded in cross-aggregation?*

To this end, this paper proposes the *Universal Graph Cross-attention Transformer (UGCFormer)*, which implements the cross-aggregation mechanism via cross-attention. To be specific, it separately encodes graph topology and node attributes to obtain their initial representations. At its core lies a linearized Dual Cross-Attention (DCA) module that updates the topology and attribute representations by computing cross-attention scores among nodes and utilizing them for weighted aggregation. In theory, the DCA module adaptively captures both correlation and exclusion relationships between graph topology and node attributes, making it *simple yet effective*. Finally, the two representations are integrated to yield a comprehensive node representation. To prevent representation distortion, a consistency constraint is introduced to enforce mutual alignment between them.

The main contributions of this work are summarized as follows:

- **Mechanism Revelation**: We theoretically reveal a unified mechanism across typical Graph Transformers, namely cross-aggregation between graph topology and node attributes.
- **Model Innovation**: We propose UGCFormer, a GT architecture equipping with a linearized Dual Cross-Attention (DCA) module that implements the cross-aggregation mechanism.
- **Comprehensive Evaluation**: Extensive evaluations conducted on sixteen homophilic, heterophilic, and large-scale graphs demonstrate the universality and scalability of UGCFormer.

## 2 Preliminaries

This section begins by presenting the notation used throughout this paper. Then, it introduces the concepts of Graph Neural Networks (GNNs) and Graph Transformers (GTs).

### 2.1 Notations

The subject of this paper is the widely-used undirected attribute graph, denoted as $\mathcal{G}(\mathcal{V}, \mathcal{E})$, where $\mathcal{V}$ and $\mathcal{E}$ represent the node set and edge set. $\mathcal{V}$ consists of $n$ node instances $\{(\mathbf{x}_v, \mathbf{y}_v)\}_{v \in \mathcal{V}}$, where

---

[1]The ability of models to handle both homophilic and heterophilic graphs.

$\mathbf{x}_v \in \mathbb{R}^f$ and $\mathbf{y}_v \in \mathbb{R}^c$ denote the node attribute and label of node $v$, respectively. $f$ is the dimension of attributes and $c$ is the dimension of labels. $\mathcal{E} = \{(v_i, v_j)\}$ terms the edge set. Typically, graph topology is described by the adjacency matrix $\mathbf{A} \in \mathbb{R}^{n \times n}$ where $a_{i,j} = 1$ only if $(v_i, v_j) \in \mathcal{E}$, and $a_{i,j} = 0$ otherwise. In formal terms, the graph $\mathcal{G}$ can be redescribed as $\mathcal{G}(\mathbf{A}, \mathbf{X})$. In the context of semi-supervised learning, the node labels are segmented into two sets: $\mathbf{Y}_L \in \mathbb{R}^{n_l \times c}$ for the labeled nodes and $\mathbf{Y}_U \in \mathbb{R}^{n_u \times c}$ for the unlabeled nodes.

To verify the model's universality, this paper examines graphs with varying degrees of homophily. In *homophilic* graphs, edges are typically formed between nodes with similar labels. Conversely, in *heterophilic* graphs, edges tend to form between nodes with dissimilar labels [37, 6, 60, 62].

## 2.2 Graph Neural Networks

Message Passing (MP)-based Graph Neural Networks (GNNs) follow an aggregation-combination strategy. Specifically, the representation of each node is iteratively updated by aggregating the features from its local neighbors and combining the aggregated features with its features, which is given by

$$\mathbf{h}_v^l \triangleq COM^l \left( \mathbf{h}_v^{l-1}, AGG^l \left( \{ \mathbf{h}_u^{l-1} | u \in \mathcal{N}(v) \} \right) \right), \tag{1}$$

where $\mathcal{N}(v)$ denotes the set of neighboring nodes of node $v$. For the functions $AGG(\cdot)$ and $COM(,)$, vanilla GNNs, *e.g.*, GCN [29], adopt the sum function to implement them, that is,

$$GCN(\mathbf{A}, \mathbf{H}): \ \mathbf{H}^{l+1} = \sigma(\tilde{\mathbf{A}} \mathbf{H}^l \mathbf{W}), \ \mathbf{H}^0 = \mathbf{X}, \tag{2}$$

where $\sigma(\cdot)$ stands for the nonlinear activation functions, and $\tilde{\mathbf{A}} = \hat{\mathbf{D}}^{-\frac{1}{2}} \hat{\mathbf{A}} \hat{\mathbf{D}}^{-\frac{1}{2}}$ is the normalized adjacency matrix with $\hat{\mathbf{A}} = \mathbf{A} + \mathbf{I}$. $\mathbf{W}$ denotes the trainable projection parameters.

## 2.3 Transformers

Inspired by the success of Transformers in NLP [46], numerous variant models have been designed for multiple fields, including CV [21] and Graph Learning. They typically consist of four functional components: *attention module, feed-forward network, residual connection, and normalization.*

**Self-attention Module.** This is a core component of the vanilla Transformer to model intra-sequence relationships among all tokens [46]. Given a sequence containing $n$ tokens $\mathbf{H} = [\mathbf{h}_i]_{i=0}^{n-1} \in \mathbb{R}^{n \times d}$, the module first projects $\mathbf{H}$ into Query $q(\mathbf{H})$, Key $k(\mathbf{H})$, and Value $k(\mathbf{H})$. It then employs the attention scores calculated from all Query-Key pairs to perform a weighted sum of the Value vectors.

A general formulation of the Self-Attention (SA) module is given by

$$SA(\mathbf{H}): \ \hat{\mathbf{H}}_{SA} = Softmax \left( \frac{q(\mathbf{H}) k(\mathbf{H})^\top}{\sqrt{d}} \right) v(\mathbf{H}), \tag{3}$$

where $q(\cdot)$, $k(\cdot)$, and $v(\cdot)$ generate the Query, Key, and Value via MLPs [41] with learnable parameters $\mathbf{W}$. The attention score $Softmax(q(\mathbf{H})k(\mathbf{H})^\top/\sqrt{d}) \in \mathbb{R}^{n \times n}$ is computed via the scaled dot product of full-token pairs, resulting in a quadratic computational complexity.

**Graph Transformers (GTs).** Most existing models [51, 1, 39, 36, 4, 53, 61, 3] build upon the SA module. GTs differ from traditional Transformers in how they leverage topology information to capture structural biases. As discussed in the Introduction, two main strategies for incorporating topology information have achieved SOTA performance on node-level tasks: (1) integrating GNN blocks, and (2) modulating node attributes utilizing Positional Encodings (PEs).

**Cross-Attention Module.** Unlike self-attention, which models the intra-source relationships, cross-attention captures the interactions between two distinct sources. For the features from two different sources $\mathbf{H} \in \mathbb{R}^{n_1 \times d}$ and $\mathbf{Z} \in \mathbb{R}^{n_2 \times d}$, the Cross-Attention (CA) module can be expressed as

$$CA(\mathbf{Z}, \mathbf{H}): \ \hat{\mathbf{H}}_{CA} = Softmax \left( \frac{q(\mathbf{Z}) k(\mathbf{H})^\top}{\sqrt{d}} \right) v(\mathbf{H}). \tag{4}$$

After the representation $\hat{\mathbf{H}}_{CA}$ is obtained, it is typically used as the cross-source representation to update $\mathbf{Z}$. Due to its exceptional capacity for modeling inter-source relationships, this module has been applied in diverse domains, *e.g.*, NLP [15] and CV [26]. However, it has received little attention in Graph Learning, largely due to *the lack of motivation and well-defined applied target.* Moreover, similar to the self-attention (Eq. 3), its computational complexity is quadratic, *i.e.*, $O(n_1 n_2)$.

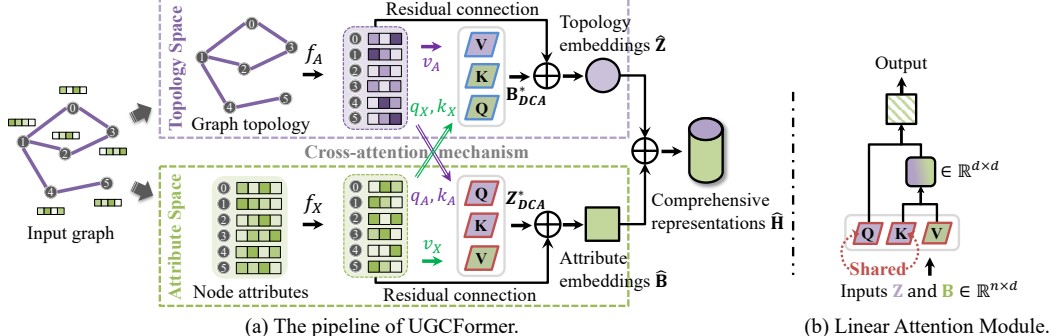

(a) The pipeline of UGCFormer.  (b) Linear Attention Module.

Figure 1: Overview of the proposed GT architecture UGCFormer and its linear attention module. (a) The pipeline of UGCFormer, which incorporates a dual cross-attention (DCA) module. First, two basic elements of graphs (*i.e.*, graph topology and node attributes) are independently processed in their respective spaces utilizing distinct projection layers $f_A(\cdot)$ and $f_X(\cdot)$. Next, the dual cross-attention (DCA) module with residual connections operates across the topology and attribute spaces, updating each representation by integrating correlated features from the other space. Finally, the two representations are combined to produce the final output representation. (b) Illustration of the proposed efficient cross-attention module, where parameters are shared between the query ($\mathbf{Q}$) and key ($\mathbf{K}$), and the representations are computed using linearized attention, given by $\mathbf{Q}(\mathbf{K}^\top \mathbf{V})$.

## 3 Methodology

This section starts by theoretically exploring the functional mechanism shared by Graph Transformers (GTs) that use Graph Neural Network (GNN) blocks and GTs that utilize Positional Encodings (PEs). Inspired by this mechanism, it introduces *UGCFormer*, a simple yet universal graph cross-attention Transformer with linear complexity. Finally, it gives a comprehensive analysis of UGCFormer.

### 3.1 Motivations

As previously discussed, the underlying mechanism behind the effectiveness of typical GTs remains insufficiently explored. To address this issue, this subsection proposes a cross-aggregation mechanism and theoretically examines how it is manifested in the two types of SOTA GTs.

The cross-aggregation mechanism is formally defined as follows.

**Definition 1.** (Cross-aggregation mechanism) Given two representations $\mathbf{B} \in \mathbb{R}^{n_1 \times d_1}$ and $\mathbf{Z} \in \mathbb{R}^{n_2 \times d_2}$ from different modalities (sources), which share at least one same dimension, *i.e.*, $n_1 = n_2$ or $d_1 = d_2$. A general formula for two types of cross-aggregations can be expressed as

$$\hat{\mathbf{Z}} \triangleq \begin{cases} Sim(\mathbf{Z}, \mathbf{B})\mathbf{B}, & \text{if } d_1 = d_2, \\ \mathbf{B}\, Sim(\mathbf{B}, \mathbf{Z}), & \text{if } n_1 = n_2, \end{cases} \tag{5}$$

where $Sim(\mathbf{Z}, \mathbf{B})$ denotes a similarity function between $\mathbf{Z}$ and $\mathbf{B}$, such as cosine similarity.

The first case corresponds to *sample (node)-level* aggregation, *e.g.*, cross-attention (Eq. 4), while the second corresponds to *dimension (feature)-level* aggregation [56, 57, 61]. When $\mathbf{Z} = \mathbf{B}$, Eq. 5 reduces to a self-aggregation (e.g., self-attention). Accordingly, two theorems are presented.

**Theorem 1.** *In typical Graph Transformers, the diffusion matrix of GNN blocks can be expressed via eigendecomposition as $\mathbf{S} = \mathbf{U}\mathbf{\Lambda}\mathbf{U}^\top$, where $\mathbf{U}$ and $\mathbf{\Lambda} = diag([\lambda_1, \dots, \lambda_n])$ represents eigenvectors and eigenvalues, respectively (in descending order). Accordingly, the GNN block can be viewed as a cross-aggregation between attribute representations $\mathbf{X}\mathbf{W}$ and topology representations $\mathbf{U}\sqrt{\mathbf{\Lambda}}$.*

**Theorem 2.** *Given the modulated node attributes using any PE, i.e., $\hat{\mathbf{X}} = [\mathbf{X}; \mathbf{P}]$, where $\mathbf{P} \in \mathbb{R}^{n \times k}$ represents the PE and $[;]$ denotes concatenation operator. PE-based GTs (Eq. 3) inherently contain a cross-aggregation between attribute representations $\mathbf{X}\mathbf{W}$ and topology representations $\mathbf{P}\mathbf{W}$.*

The proofs for Theorems 1 and 2 are provided in Sections B and C, respectively. In short, the key mechanism of GTs using GNNs and PEs is *Cross-Aggregation between topology and attributes*.

## 3.2 UGCFormer

Motivated by the cross-aggregation mechanism explored in the previous subsection, this subsection introduces UGCFormer, a simple yet universal GT. At its core, UGCFormer employs a linearized cross-attention module that implements the cross-aggregation mechanism to capture interactions between graph topology and node attributes. UGCFormer consists of four modules, each of which is described below. The detailed implementation is provided in Algorithm 1.

**Initial Representation Layer.** Two different projection layers are utilized to independently generate initial representations for the two types of graph information. For simplicity, MLPs are used to process the adjacency matrix $\mathbf{A} \in \mathbb{R}^{n \times n}$ and the attribute matrix $\mathbf{X} \in \mathbb{R}^{n \times f}$, producing the corresponding initial representations $\mathbf{Z}$ and $\mathbf{B}$, that is,

$$\mathbf{Z}^0 = MLP_A(\mathbf{A}), \quad \mathbf{B}^0 = MLP_X(\mathbf{X}) \in \mathbb{R}^{n \times d}, \tag{6}$$

where $MLP_A(\cdot)$ and $MLP_X(\cdot)$ term the MLPs for processing topology and attributes, respectively.

**Dual Cross-attention Module.** As an implementation of the cross-aggregation (in Definition 1), this module is designed to capture the interactions between these two types of graph information. However, directly employing the cross-attention (Eq. 4) may result in two issues: (1) unacceptable quadratic computational complexity due to the calculation of dot products for all node pairs, and (2) an increased number of parameters and overfitting risk due to the use of two separate channels.

To alleviate these drawbacks, the proposed Dual Cross-Attention (DCA) module adopts two strategies: (1) *linearized attention computation* [50] and (2) *parameter sharing*, as shown in Fig. 1(b). Firstly, through approximating or replacing the Softmax attention utilizing separate kernel functions, the computation order in the SA module can be reordered from the standard (Query×Key)×Value (Eq. 3) to the more efficient Query×(Key×Value) format [28]. However, this strategy is unsuitable for the cross-attention module. Specifically, the attention score $k(\mathbf{B})^\top v(\mathbf{B}) \in \mathbb{R}^{d \times d}$ computes the similarity between features within the same space, rather than across different spaces. To ensure cross-space interaction, DCA sets the Key to originate from the same space as the Query. Moreover, DCA shares parameters between the Query and Key to reduce the number of parameters.

To streamline the description of the update process for topology representations and attribute representations, two abstract representations $\mathbf{H}_1$ and $\mathbf{H}_2$ are introduced. For clarity, layer indices are omitted. The general formulation of the DCA module is given as follows:

$$DCA(\mathbf{H}_1, \mathbf{H}_2): \quad \mathbf{Q} = q(\mathbf{H}_1), \mathbf{K} = k(\mathbf{H}_1), \mathbf{V} = v(\mathbf{H}_2), \tag{7}$$

$$\tilde{\mathbf{Q}} = \frac{\mathbf{Q}}{\|\mathbf{Q}\|_{\mathcal{F}}}, \quad \tilde{\mathbf{K}} = \frac{\mathbf{K}}{\|\mathbf{K}\|_{\mathcal{F}}}, \tag{8}$$

$$\mathbf{H}_1^* = \mathbf{D}^{-1} \left( \mathbf{V} + \frac{1}{n} \tilde{\mathbf{Q}} (\tilde{\mathbf{K}}^\top \mathbf{V}) \right), \tag{9}$$

where $q(\cdot)$ and $k(\cdot)$ stand for the Query and Key functions, respectively, with $q(\cdot) = k(\cdot)$. And $v_A(\cdot)$ represents the Value function. These functions are implemented as MLPs. $\| \cdot \|_{\mathcal{F}}$ denotes the Frobenius norm. $\mathbf{D} = Diag(\mathbf{1} + \frac{1}{n} \tilde{\mathbf{Q}} (\tilde{\mathbf{K}}^\top \mathbf{1}))$ stands for a diagonal matrix and $\mathbf{1}$ is an all-one vector.

The topology-related attribute representations can be obtained as $\mathbf{B}_{DCA}^* = DCA(\mathbf{B}, \mathbf{Z})$. Then, the topology representations are updated via

$$\hat{\mathbf{Z}} = (1 - \tilde{\lambda}) \tilde{\mathbf{A}} \mathbf{V} + \tilde{\lambda} \mathbf{B}_{DCA}^*, \tag{10}$$

where $\tilde{\mathbf{A}}$ denotes the normalized adjacency matrix. The first term denotes the topology representation updated purely from the topology space, which can be viewed as being obtained via *spectral clustering* [48, 54] (see Theorem 3). $\tilde{\lambda} = \mathrm{Tanh}(\lambda)$ stands for a scalar to balance these two terms, where $\lambda$ is a learnable parameter. Combining these two terms allows for the fusion of topological details alongside the topology-related attribute information into the final topology representations.

Similarly, the attribute representations are updated by incorporating relevant information from the topology space, that is, $\mathbf{Z}_{DCA}^* = DCA(\mathbf{Z}, \mathbf{B})$, with their representations. This can be expressed as

$$\hat{\mathbf{B}} = (1 - \tilde{\gamma}) \mathbf{B}^0 + \tilde{\gamma} \mathbf{Z}_{DCA}^*, \tag{11}$$

where $\tilde{\gamma} = \mathrm{Tanh}(\gamma)$ denotes a scalar to trade off the two terms with $\gamma$ denotes a learnable parameter. Note that DCA requires two separate sets of network parameters to generate the attribute and topology

**Algorithm 1:** UGCFormer

---

**Input:** Graph $\mathcal{G}(\mathbf{A}, \mathbf{X})$ with labels $\mathbf{Y}$, hyperparameters $\alpha$, $\beta$ and $\tau$.
**Output:** Trained network parameters $\Theta^*$.
**Initialization:** Network parameters $\Theta$,
**while** not converged **do**
> 1. Generate two initial node representations $\mathbf{Z}^0$ and $\mathbf{B}^0$ via Eq. 6;
> 2. Get two updated node representations $\hat{\mathbf{Z}}$ and $\hat{\mathbf{B}}$ via Eqs. 10 and 11;
> 3. Obtain the final predictions $\hat{\mathbf{Y}}$ via Eq. 12;
> 4. Calculate the overall loss $\mathcal{L}_{final}$ via Eqs. 13 and 14;
> 5. Optimize the parameters via $\Theta^* \leftarrow \text{Adam}(\mathcal{L}, \Theta)$;

**end**
**return** Parameters $\Theta^*$

---

representations, *e.g.*, $q_A(\cdot)$ and $q_X(\cdot)$ represent the Query for graph topology and node attributes, respectively, as shown in Fig. 1.

**Prediction Layer.** After obtaining the topology representations $\hat{\mathbf{Z}}$ and attribute representations $\hat{\mathbf{B}}$ through $l$ layers, the final node representations can be generated by weight combining them. Next, the predictions are generated via an MLP network and nonlinearities (*i.e.*, $Softmax(\cdot)$), that is,

$$\hat{\mathbf{Y}} = Softmax \left( MLP \left( (1-\alpha)\hat{\mathbf{Z}} + \alpha\hat{\mathbf{B}} \right) \right), \tag{12}$$

where $\alpha$ denotes a scalar that adjusts attention to topology and attribute representations. $\hat{\mathbf{Y}} \in \mathbb{R}^{n \times c}$ represents the predictions, indicating the estimated outcomes for each of the $n$ nodes across $c$ classes.

**Objective Function.** Note that the proposed DCA module, with a large number of parameters across two distinct spaces, is susceptible to representation distortion caused by overfitting [8], especially when the number of training nodes is limited. Thus, a consistency constraint is introduced to align the two representations $\hat{\mathbf{Z}}$ and $\hat{\mathbf{B}}$. First, pseudo-labels are derived by averaging the two representations. For a node $v$, its pseudo-label can be computed as $\mathbf{y}_v = \frac{1}{2}(\hat{\mathbf{z}}_v + \hat{\mathbf{b}}_v)$. Next, low-entropy pseudo-labels are obtained through a sharpening technique that controls the sharpness of the distribution. This can be formulated as $\bar{y}_{i,j} = y_{i,j}^{\frac{1}{\tau}} / \sum_{k=0}^{c-1} y_{i,j}, (0 \leq j \leq c-1)$, where $\tau \in (0,1]$ denotes a scaling factor that controls the sharpness of the distribution.

Once the pseudo-label is obtained, the next step is to calculate the squared Euclidean distance between it and the two representations, which is given by

$$\mathcal{L}_{con}(\hat{\mathbf{Z}}, \hat{\mathbf{B}}) = \frac{1}{2} \sum_{i}^{n-1} \left( \|\bar{\mathbf{y}}_i - \hat{\mathbf{z}}_i\|_2^2 + \|\bar{\mathbf{y}}_i - \hat{\mathbf{b}}_i\|_2^2 \right). \tag{13}$$

The overall objective of UGCFormer is to minimize the weighted sum of the cross-entropy loss and the consistency loss, defined as follows:

$$\mathcal{L}_{overall} = \mathcal{L}_{ce} + \beta\mathcal{L}_{con}, \tag{14}$$

where $\mathcal{L}_{ce} = -\sum_{v \in \mathcal{V}_L} \mathbf{y}_v \log \hat{\mathbf{y}}_v$ and $\beta$ stands for a balance hyperparameter.

### 3.3 Model Analysis

This subsection provides a comprehensive analysis of UGCFormer. First, the computational complexity of UGCFormer is analyzed. Then, the simplicity of UGCFormer is examined through architectural comparison with existing GTs. Finally, the effectiveness of UGCFormer is theoretically justified.

**Complexity Analysis.** UGCFormer operates with **linear time complexity**. The time complexity for generating initial representations through the projection layer is $O(md+nd^2)$ as the adjacency matrix is sparse, where $m$ represents the number of edges. Secondly, owing to the linearized cross-attention module, the aggregation operator incurs a computational overhead of $O(nd^2)$. Finally, obtaining the predictions involves feature mapping and element-wise operations, resulting in a complexity of $O(nd)$. UGCFormer operates with **linear space complexity**. The space required to store the input topology

Table 1: Accuracy (ACC) or ROC-AUC in percentage (mean$_{\pm\text{std}}$) over 10 trials of the node classification task on homophilic graphs. Best and runner-up models are in bold and underlined, respectively.

| Model
Metric | Cora
ACC ↑ | CiteSeer
ACC ↑ | PubMed
ACC ↑ | Photo
ACC ↑ | CS
ACC ↑ | Physics
ACC ↑ | Questions
ROC-AUC ↑ | Avg ↑ | Rank ↓ |
|---|---|---|---|---|---|---|---|---|---|
| GCN | $81.60_{\pm0.40}$ | $71.60_{\pm0.40}$ | $78.80_{\pm0.60}$ | $92.70_{\pm0.20}$ | $92.92_{\pm0.12}$ | $96.18_{\pm0.07}$ | $76.28_{\pm0.64}$ | 84.30 | 13.29 |
| GAT | $83.00_{\pm0.70}$ | $72.10_{\pm1.10}$ | $79.00_{\pm0.40}$ | $93.87_{\pm0.10}$ | $93.61_{\pm0.14}$ | $96.17_{\pm0.08}$ | $74.94_{\pm0.56}$ | 84.67 | 11.43 |
| GraphSAGE | $82.68_{\pm0.47}$ | $71.93_{\pm0.85}$ | $79.41_{\pm0.53}$ | $94.59_{\pm0.14}$ | $93.91_{\pm0.13}$ | $96.49_{\pm0.06}$ | $76.44_{\pm0.62}$ | 85.06 | 9.71 |
| APPNP | $83.30_{\pm0.50}$ | $71.80_{\pm0.50}$ | $80.10_{\pm0.20}$ | $94.32_{\pm0.14}$ | $94.49_{\pm0.07}$ | $96.54_{\pm0.07}$ | $75.51_{\pm0.23}$ | 85.15 | 8.14 |
| GPR-GNN | $84.20_{\pm0.50}$ | $71.60_{\pm0.80}$ | $80.07_{\pm0.92}$ | $94.49_{\pm0.16}$ | $95.13_{\pm0.09}$ | $96.85_{\pm0.02}$ | $67.15_{\pm1.92}$ | 84.21 | 8.71 |
| LINKX | $77.95_{\pm0.12}$ | $68.25_{\pm0.24}$ | $77.36_{\pm0.42}$ | $91.97_{\pm0.19}$ | $94.77_{\pm0.19}$ | $96.29_{\pm0.13}$ | $75.71_{\pm1.40}$ | 83.19 | 13.14 |
| GloGNN | $82.17_{\pm0.29}$ | $71.74_{\pm0.88}$ | $\underline{80.37}_{\pm0.95}$ | $95.10_{\pm0.20}$ | $95.00_{\pm0.10}$ | $96.97_{\pm0.15}$ | $67.15_{\pm1.92}$ | 84.07 | 8.43 |
| GraphGPS | $82.84_{\pm1.03}$ | $72.73_{\pm1.23}$ | $79.94_{\pm0.26}$ | $95.06_{\pm0.13}$ | $93.93_{\pm0.12}$ | $97.12_{\pm0.19}$ | $71.73_{\pm1.47}$ | 84.76 | 8.29 |
| NodeFormer | $82.20_{\pm0.90}$ | $72.50_{\pm1.10}$ | $79.90_{\pm1.00}$ | $93.46_{\pm0.35}$ | $95.64_{\pm0.22}$ | $96.24_{\pm0.24}$ | $74.27_{\pm1.46}$ | 84.89 | 9.57 |
| NAGphormer | $82.12_{\pm1.18}$ | $71.47_{\pm1.30}$ | $79.73_{\pm0.28}$ | $95.49_{\pm0.11}$ | $\underline{95.75}_{\pm0.09}$ | $\underline{97.34}_{\pm0.03}$ | $74.98_{\pm0.63}$ | 85.27 | 7.71 |
| Exphormer | $82.77_{\pm1.38}$ | $71.63_{\pm1.19}$ | $79.46_{\pm0.35}$ | $95.35_{\pm0.22}$ | $94.93_{\pm0.01}$ | $96.89_{\pm0.09}$ | $74.67_{\pm0.79}$ | 85.10 | 8.86 |
| GOAT | $83.18_{\pm1.27}$ | $71.99_{\pm1.26}$ | $79.13_{\pm0.38}$ | $92.96_{\pm1.48}$ | $94.21_{\pm0.38}$ | $96.45_{\pm0.28}$ | $75.76_{\pm1.66}$ | 84.81 | 10.00 |
| SGFormer | $\underline{84.50}_{\pm0.80}$ | $72.60_{\pm0.20}$ | $80.30_{\pm0.60}$ | $95.10_{\pm0.47}$ | $94.78_{\pm0.20}$ | $96.60_{\pm0.18}$ | $72.15_{\pm1.31}$ | 85.14 | 6.57 |
| Polynormer | $83.25_{\pm0.93}$ | $72.31_{\pm0.78}$ | $79.24_{\pm0.43}$ | $\mathbf{96.46}_{\pm0.26}$ | $95.53_{\pm0.16}$ | $97.27_{\pm0.08}$ | $\underline{76.91}_{\pm1.63}$ | $\underline{85.85}$ | $\underline{4.71}$ |
| Gradformer | $82.95_{\pm0.73}$ | $\underline{72.80}_{\pm0.59}$ | $80.14_{\pm0.48}$ | $95.76_{\pm0.28}$ | $94.21_{\pm0.29}$ | $97.06_{\pm0.16}$ | $74.71_{\pm1.07}$ | 85.38 | 6.14 |
| UGCFormer | $\mathbf{84.94}_{\pm0.43}$ | $\mathbf{73.41}_{\pm0.27}$ | $\mathbf{81.79}_{\pm0.81}$ | $\underline{96.21}_{\pm0.31}$ | $\mathbf{95.91}_{\pm0.23}$ | $\mathbf{97.35}_{\pm0.17}$ | $\mathbf{77.02}_{\pm0.76}$ | $\mathbf{86.66}$ | $\mathbf{1.14}$ |

and attributes is $O(m + nd)$, where $m$ corresponds to the number of edges and $nd$ accounts for the feature matrix. The aggregated and updated representations each require $O(nd)$ space, since their dimensions do not exceed those of the input feature matrix. In the linearized attention computation (Fig. 1(b)), the attention matrix contributes an additional $O(d^2)$ space overhead.

**Components.** To leverage discriminative graph topology and capture structural biases, existing GTs often resort to auxiliary components that compromise their efficiency and effectiveness. Specifically, the positional or structural encodings (*e.g.*, Laplacian eigenvector encodings) used in GraphGPS [39], NAGphormer [2], Exphormer [44], and GOAT [31] as well as augmented training losses (*e.g.*, edge regularization loss) in NodeFormer, often necessitate cubic computational complexity and quadratic space consumption. Moreover, the GNN module tends to generate representations that are susceptible to issues caused by the limited message passing. In contrast, the proposed UGCFormer features a streamlined and efficient design that relies solely on a linear cross-attention module.

**Theoretical Justification.** Though designed to be simple and intuitive, the proposed UGCFormer is theoretically guaranteed to be effective from a graph optimization perspective [55, 58].

**Theorem 3.** *Let $\mathbf{Z}$ and $\mathbf{B}$ denote the topology representations and attribute representations, respectively. The representation update in the dual cross-attention module DCA (Eq. 10 and Eq. 11) is equivalent to solving an optimization problem with the objective function:*

$$\underset{\mathbf{Z},\mathbf{B}}{\arg\min}\, \lambda \operatorname{Tr}(\mathbf{Z}^\top \tilde{\mathbf{L}}\mathbf{Z}) + \|\mathbf{B} - MLP(\mathbf{X})\|_F^2 - \eta\|\mathbf{Z}^\top \mathbf{B}\|_F^2, \qquad (15)$$

*where $\tilde{\mathbf{L}}$ terms the Laplacian matrix of $\tilde{\mathbf{A}}$, $\lambda$ and $\eta$ are the scalars used to balance these three terms.*

In Eq. 15, the first term stands for a relaxed optimization problem widely used in spectral clustering [48]. Thus, the DCA seeks to generate topology representations that capture mesoscopic community structures. The second term measures the distance between the attribute representation $\mathbf{B}$ and its initial representation $MLP(\mathbf{X})$. The third term denotes the statistical dependence measure, approximated by the Hilbert-Schmidt Independence Criterion (HSIC) [19], that is, $\text{HSIC}(\mathbf{Z}, \mathbf{B}) \approx \operatorname{Tr}(\mathbf{Z}\mathbf{Z}^\top \mathbf{B}\mathbf{B}^\top) = \|\mathbf{Z}^\top \mathbf{B}\|_F^2$, which reflects the dependence between topology and attribute representations. Therefore, the interaction, whether mutual correlation (positive weights) or exclusion (negative weights), can be modulated by the parameters. In summary, Theorem 3 indicates that UGCFormer focuses on learning representations by mining the interactions of two basic graph information.

## 4 Experiments

This section evaluates the effectiveness and universality of the proposed UGCFormer by comparing its performances against various diverse graph learning models on the node classification task. Moreover,

Table 2: Accuracy (ACC) in percentage (mean$_{\pm\text{std}}$) over 10 trials of the node classification task on heterophilic graphs. Best and runner-up models are in bold and underlined, respectively.

| Model
Metric | Cornell
ACC ↑ | Texas
ACC ↑ | Wisconsin
ACC ↑ | Actor
ACC ↑ | Chameleon
ACC ↑ | Squirrel
ACC ↑ | Ratings
ACC ↑ | Avg ↑ | Rank ↓ |
|---|---|---|---|---|---|---|---|---|---|
| GCN | $58.41_{\pm3.28}$ | $65.61_{\pm4.80}$ | $61.28_{\pm5.87}$ | $30.63_{\pm0.62}$ | $43.43_{\pm1.92}$ | $41.30_{\pm0.94}$ | $47.77_{\pm0.69}$ | 49.78 | 11.57 |
| GAT | $58.29_{\pm3.52}$ | $60.73_{\pm6.20}$ | $63.64_{\pm6.18}$ | $30.36_{\pm0.94}$ | $40.14_{\pm1.57}$ | $35.09_{\pm0.70}$ | $47.95_{\pm0.53}$ | 48.03 | 14.57 |
| GraphSAGE | $75.95_{\pm5.31}$ | $82.43_{\pm6.07}$ | $81.18_{\pm4.56}$ | $34.23_{\pm1.07}$ | $39.11_{\pm5.05}$ | $36.46_{\pm2.16}$ | $53.11_{\pm0.54}$ | 57.50 | 11.00 |
| APPNP | $73.68_{\pm3.97}$ | $74.57_{\pm2.48}$ | $70.61_{\pm3.47}$ | $35.18_{\pm1.21}$ | $39.42_{\pm3.87}$ | $38.13_{\pm2.67}$ | $49.78_{\pm0.72}$ | 54.49 | 12.71 |
| GPR-GNN | $78.11_{\pm6.55}$ | $81.35_{\pm5.32}$ | $82.55_{\pm6.23}$ | $35.16_{\pm0.85}$ | $39.93_{\pm3.30}$ | $38.95_{\pm1.99}$ | $43.90_{\pm0.48}$ | 57.14 | 11.86 |
| LINKX | $77.84_{\pm5.81}$ | $74.60_{\pm8.37}$ | $75.49_{\pm5.72}$ | $36.10_{\pm1.55}$ | $40.02_{\pm2.35}$ | $39.88_{\pm2.53}$ | $51.36_{\pm0.47}$ | 56.47 | 11.00 |
| GloGNN | $\underline{83.51}_{\pm4.26}$ | $\underline{84.32}_{\pm4.15}$ | $\underline{87.06}_{\pm3.53}$ | $37.35_{\pm1.30}$ | $38.43_{\pm3.74}$ | $30.30_{\pm1.92}$ | $37.28_{\pm0.66}$ | 56.89 | 8.14 |
| GraphGPS | $82.06_{\pm5.73}$ | $82.21_{\pm6.14}$ | $85.36_{\pm4.24}$ | $36.18_{\pm1.27}$ | $40.79_{\pm4.03}$ | $39.67_{\pm2.84}$ | $53.10_{\pm0.42}$ | 59.91 | 7.29 |
| NodeFormer | $82.15_{\pm6.72}$ | $81.68_{\pm4.65}$ | $83.41_{\pm5.51}$ | $36.28_{\pm1.25}$ | $43.09_{\pm2.81}$ | $40.61_{\pm1.25}$ | $50.12_{\pm0.64}$ | 59.62 | 7.43 |
| NAGphormer | $79.97_{\pm6.07}$ | $80.18_{\pm4.57}$ | $82.97_{\pm2.98}$ | $34.36_{\pm0.75}$ | $\underline{44.61}_{\pm3.10}$ | $41.27_{\pm1.09}$ | $52.51_{\pm0.83}$ | 59.41 | 7.86 |
| Exphormer | $83.07_{\pm4.31}$ | $82.81_{\pm3.52}$ | $83.90_{\pm4.31}$ | $36.82_{\pm1.95}$ | $41.63_{\pm3.12}$ | $40.32_{\pm1.59}$ | $52.08_{\pm0.81}$ | 60.06 | 6.00 |
| GOAT | $83.18_{\pm1.27}$ | $71.99_{\pm1.26}$ | $79.13_{\pm0.38}$ | $36.55_{\pm1.19}$ | $42.56_{\pm3.17}$ | $40.81_{\pm0.54}$ | $49.68_{\pm0.50}$ | 57.70 | 8.53 |
| SGFormer | $81.64_{\pm3.88}$ | $84.29_{\pm5.67}$ | $83.59_{\pm5.42}$ | $\mathbf{37.79}_{\pm1.89}$ | $\mathbf{44.93}_{\pm3.91}$ | $\mathbf{41.80}_{\pm2.27}$ | $48.01_{\pm0.49}$ | $\underline{60.29}$ | $\underline{4.86}$ |
| Polynormer | $81.90_{\pm4.17}$ | $82.57_{\pm5.11}$ | $83.95_{\pm2.98}$ | $37.01_{\pm1.10}$ | $41.97_{\pm3.18}$ | $40.87_{\pm1.96}$ | $\underline{53.29}_{\pm0.23}$ | 60.22 | 5.14 |
| Gradformer | $83.06_{\pm5.16}$ | $82.19_{\pm5.24}$ | $84.26_{\pm2.24}$ | $36.58_{\pm0.71}$ | $40.73_{\pm3.69}$ | $40.29_{\pm1.88}$ | $53.11_{\pm0.29}$ | 60.03 | 6.43 |
| UGCFormer | $\mathbf{85.14}_{\pm5.83}$ | $\mathbf{84.59}_{\pm4.69}$ | $\mathbf{87.36}_{\pm3.30}$ | $\underline{37.41}_{\pm0.79}$ | $43.28_{\pm2.17}$ | $\underline{41.56}_{\pm2.01}$ | $\mathbf{53.48}_{\pm0.14}$ | $\mathbf{61.83}$ | $\mathbf{1.71}$ |

it provides additional analysis experiments to enhance the understanding of UGCFormer. Refer to Section E for details on the datasets, baselines, and experimental setups.

## 4.1 Experimental Results

**Homophilic Graphs.** The experiment results for node classification on homophilic graphs are shown in Tab. 1, from which three key observations can be made. Firstly, the performance of the backbone GNNs (*e.g.*, GCN and GAT) lags behind that of GTs. To be specific, on six of the seven homophilic graphs, the models that rank in the top two positions are GTs. This is primarily because most GTs, such as NAGphormer, are built upon these backbone GNNs and specifically address the shortcomings of GNNs in capturing long-range dependencies. Secondly, the proposed UGCFormer outperforms all baseline GTs across six of the seven datasets and achieves the optimal rank, demonstrating its consistent superior performance. In particular, on PubMed, UGCFormer achieves a performance that is $2.55\%$ higher than the baseline Polynormer, which has an average rank of second, and its average rank is significantly lower. Thirdly, compared with the baseline LINKX, which also processes graph topology and node attributes separately and does not leverage message passing, UGCFormer consistently achieves better results across all datasets. This can be attributed to its ability to capture the interactions between these two types of graph information and alleviate the representation distortion, which LINKX does not account for. This highlights the rationality of UGCFormer's design.

**Heterophilic Graphs.** Tab. 2 shows the results of the node classification task on seven heterophilic graphs, highlighting three key observations. Firstly, the baseline GTs perform slightly better than the baseline GNNs, but the difference is not substantial. In specific, the baseline GNNs, particularly GloGNN on Cornell, Texas, and Wisconsin, and GraphSAGE on the Ratings, achieve top-two results on five of the seven datasets. This can be attributed to the high complexity and large number of parameters in GTs, which make them prone to overfitting. Therefore, the baseline SGFormer, which linearly combines the local representation from the GNN module and the global representation from the GT module, achieves superior performance. This is evidenced by its ranking in the top two for three datasets. Secondly, the proposed UGCFormer outperforms the GT baselines on the majority of heterophilic graphs, proving its effectiveness. For example, on Cornell, UGCFormer exceeds the second-ranked GT, *i.e.*, GOAT, by a significant margin of $1.96\%$. Thirdly, UGCFormer consistently outperforms the baseline LINKX on all heterophilic datasets, highlighting the significance of capturing the relevance between graph topology and node attributes. Overall, UGCFormer achieves performance improvements on both homophilic and heterophilic graphs, demonstrating its universality.

**Scalability Study.** To evaluate the scalability of the proposed UGCFormer, this experiment quantitatively changes the network size and records the running time and GPU memory usage. Specifically, it utilizes the ogbn-arxiv to randomly sample subsets of nodes, with the node numbers varying from 10K to 100K. As shown in Fig. 2, the running time and GPU memory usage of UGCFormer increase linearly with the size of the sampled graph. For example, the training time and memory usage with

100k nodes are approximately five times higher than with 20k nodes. This indicates that UGCFormer exhibits linear time and space complexity, consistent with the conclusion in Section 3.3.

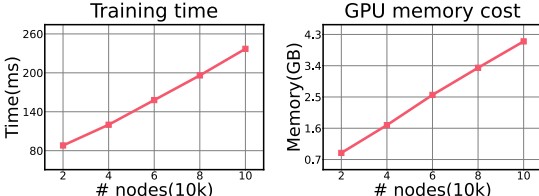

Figure 2: Training time and GPU memory usage of UGCFormer.

**Node Property Prediction.** This experiment seeks to evaluate the effectiveness and scalability of GTs by comparing them with GNNs on two large-scale benchmark datasets. Upon examining Tab. 3, which presents the results of the node property prediction task on these two datasets, two key conclusions can be drawn. Firstly, the backbone GTs generally outperform the backbone GNNs, which not only highlights the superiority of GTs but also underscores their scalability—a key challenge that GTs aim to address. This can be attributed to the integration of the GNN blocks in GTs, exemplified by SGFormer. These GTs generate the final prediction by combining the local representations from the GNN module with the global representations from the GT module. Secondly, the proposed UGCFormer achieves optimal performance on these two datasets, indicating its effectiveness and scalability on large graphs.

Table 3: Node property prediction performances on two large-scale graphs.

| Model
Metric | ogbn-proteins
ROC-AUC ↑ | ogbn-arxiv
ACC ↑ |
|---|---|---|
| GCN | $72.51_{\pm 0.35}$ | $71.74_{\pm 0.29}$ |
| GAT | $72.02_{\pm 0.44}$ | $71.95_{\pm 0.36}$ |
| GPRGNN | $71.10_{\pm 0.12}$ | $71.10_{\pm 0.12}$ |
| LINKX | $66.18_{\pm 0.33}$ | $71.59_{\pm 0.71}$ |
| GraphGPS | $76.83_{\pm 0.26}$ | $70.97_{\pm 0.41}$ |
| NodeFormer | $77.45_{\pm 1.15}$ | $67.19_{\pm 0.83}$ |
| NAGphormer | $73.61_{\pm 0.33}$ | $70.13_{\pm 0.55}$ |
| Exphormer | $74.58_{\pm 0.26}$ | $72.44_{\pm 0.28}$ |
| GOAT | $74.84_{\pm 1.16}$ | $72.41_{\pm 0.40}$ |
| SGFormer | $\underline{79.53}_{\pm 0.38}$ | $72.63_{\pm 0.13}$ |
| Polynormer | $78.97_{\pm 0.47}$ | $\underline{73.46}_{\pm 0.16}$ |
| Gradformer | $77.64_{\pm 0.51}$ | $72.71_{\pm 0.20}$ |
| UGCFormer | $\mathbf{79.95}_{\pm 0.75}$ | $\mathbf{74.02}_{\pm 0.17}$ |

## 4.2 Additional Analysis

**Ablation Study.** This experiment evaluates the contributions of the proposed cross-attention module and the consistency constraint by comparing UGCFormer with two variants lacking these components. Fig. 3 shows that these variants consistently underperform UGCFormer across the four datasets. This illustrates that the efficacy of UGCFormer stems from the collective contribution of all components. Besides, even without the consistency loss, the variant model (w/o $\mathcal{L}_{con}$) still provides competitive performance compared to the baseline GTs, as seen in Table 1. This highlights the effectiveness of the cross-attention module and thereby reaffirms the rationality of the UGCFormer architecture.

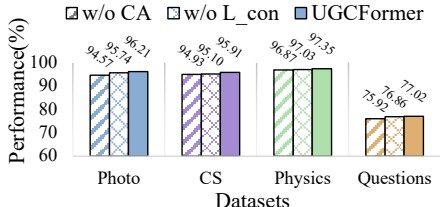

Figure 3: Impact of functional components (*i.e.*, the CA and consistency constraint).

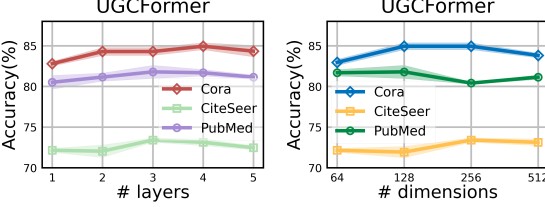

Figure 4: Performance variations for varying $l$.

Figure 5: Performance variations for varying $d$.

**Parameter Sensitivity Analysis.** These experiments aim to provide an intuitive understanding for the selection of hyperparameters. Performance changes due to varying the number of layers ($l$) and layer dimensions ($d$) are shown in Figs. 4 and 5, respectively. **Number of Layers.** Fig. 4 shows that UGCFormer achieves stable performance across various layer numbers $\{1, 2, 3, 4, 5\}$. Specifically, performance fluctuations are minimal, within $2.2\%$ on the Cora, $1.4\%$ on the CiteSeer, and $1.3\%$ on PubMed. This indicates that UGCFormer is relatively insensitive to the number of layers. Additionally, optimal performance is achieved with $\{3, 4\}$, likely due to the risk of over-smoothing in

deeper models. **Hidden Layer Dimension.** As shown in Fig. 5, UGCFormer maintains consistent performance across the hidden dimension range $\{64, 128, 256, 512\}$. For example, on the Cora, which shows the most significant performance variation, the difference is less than $2\%$. This indicates that UGCFormer is not sensitive to this parameter. Additionally, optimal performance on the three datasets corresponds to $d \in \{128, 256\}$, rather than the highest value of $512$. This suggests that larger dimensions can lead to overfitting and distorted representations. Additional hyper-parameters (including $\alpha$ and $\beta$) are analyzed in Section E.4.

## 5 Conclusions

By revisiting two typical Graph Transformers (GTs), this study has uncovered a potential functional mechanism: cross-aggregation between graph topology and node attributes. To effectively implement this mechanism, this paper introduces UGCFormer, a linearized graph cross-attention Transformer. Extensive experiments on sixteen graph benchmarks demonstrate its effectiveness and efficiency.

## 6 Acknowledgements

This work was supported in part by the National Natural Science Foundation of China (No. 92570118, U22B2036, 62376088, 62272020, 62025604, 92370111, 62272340, 62261136549), in part by the Hebei Natural Science Foundation (No. F2024202047), in part by the National Science Fund for Distinguished Young Scholarship (No. 62025602), in part by the Hebei Yanzhao Golden Platform Talent Gathering Programme Core Talent Project (Education Platform) (HJZD202509), in part by the Post-graduate's Innovation Fund Project of Hebei Province (CXZZBS2025036), in part by the Tencent Foundation, and in part by the XPLORER PRIZE.

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

# A  Algorithm Description

A layer of the proposed dual cross-attention module DCA is depicted in Algorithm 2.

---
**Algorithm 2:** PyTorch-style Code for DCA layer

---
```
# N: instance number
# D: hidden dimension
# z: data embeddings sized [N, D]
# b: data embeddings sized [N, D]
# H: head number
# Wq, Wk, Wv: parameter matrices for feature transformation
q = Wq(z) # [N, H, D]
k = Wk(z) # [N, H, D]
v = Wv(b) # [N, H, D]
# numerator
kv = torch.einsum("lhm, lhd → hmd", k, v)
num = torch.einsum("nhm, hmd → nhd", q, kv)
num += N * v # [N, H, D]
# denominator
all_ones = torch.ones(N)
k_sum = torch.einsum("lhm, l → hm", k, all_ones)
den = torch.einsum("nhm, hm → nh", q, k_sum) # [N, H]
# aggregated results
den += torch.ones_like(den) * N
output = num / den.unsqueeze(2) # [N, H, D]
# head average
output = output.mean(dim=1) # [N, D]
```
---

# B  Proof for Theorem 1

*Proof.* The proof unfolds in three stages: firstly, a concise description of the model to be validated is provided; subsequently, the model is decomposed into its topology and attribute components; and finally, it is verified that this structure aligns with the cross-aggregation form.

For a single-layer GNN block, such as GCN [29] widely used in GTs [39, 2, 50], the feature update can be expressed as

$$\mathbf{H} = \sigma(\tilde{\mathbf{A}}\mathbf{X}\mathbf{W}). \tag{16}$$

where the diffusion matrix $\tilde{\mathbf{A}}$ denotes the normalized adjacency matrix.

By omitting the nonlinear activation function and performing eigendecomposition on the adjacency matrix, the above equation can be reformulated as

$$\begin{aligned}
\mathbf{H} &= \tilde{\mathbf{A}}\mathbf{X}\mathbf{W} \\
&= \mathbf{U}\mathbf{\Lambda}\mathbf{U}^\top\mathbf{X}\mathbf{W} \\
&= \mathbf{U}\sqrt{\mathbf{\Lambda}}\sqrt{\mathbf{\Lambda}}\mathbf{U}^\top\mathbf{X}\mathbf{W}.
\end{aligned} \tag{17}$$

Next, let $\mathbf{B} = \mathbf{U}\sqrt{\mathbf{\Lambda}}$ and $\mathbf{Z} = \mathbf{X}\mathbf{W}$. The similarity function $\mathrm{Sim}(\cdot, \cdot)$ is implemented as matrix multiplication. With the definitions in Eq. 5, the GCN block can be interpreted as a cross-aggregation from the attribute space to the topology space.

**Extension to Multi-layer GNN Blocks.** Following the discussion on single-layer GNNs, the solution for multi-layer GNNs is presented. Under the above assumptions, the node representations in the $l$-th layer can be formulated as

$$\begin{aligned}
\mathbf{H}^l &= (\tilde{\mathbf{A}}^1\tilde{\mathbf{A}}^2\ldots\tilde{\mathbf{A}}^l)\mathbf{X}(\mathbf{W}^1\mathbf{W}^2\ldots\mathbf{W}^l) \\
&= \mathcal{S}\,\mathbf{X}\,\mathcal{W},
\end{aligned} \tag{18}$$

where $\tilde{\mathbf{A}}^i$ and $\mathbf{W}^i$ denote the diffusion matrix and the parameter matrix, respectively, in the $i$-th layer. $\mathcal{S} = \prod_{i=0}^{l} \mathbf{S}^l$ and $\mathcal{W} = \prod_{i=0}^{l} \mathbf{W}^l$ stand for the product of the diffusion matrices and the projection matrices. Given the properties of the diffusion matrix, the product diffusion matrix can be eigen-decomposed as $\mathcal{S} = \mathbf{U}\mathbf{\Lambda}^{(l)}\mathbf{U}^{\top}$, where $\mathbf{\Lambda}^{(l)}$ represents the $l$ power of $\mathbf{\Lambda}$. Therefore, the above conclusion still holds in the context of multi-layer GNNs. $\square$

**Remark.** This interpretation holds under the assumption that the diffusion operators across layers share a common eigenspace (*i.e.*, identical or mutually commutative $\tilde{\mathbf{A}}$ across layers). Otherwise, the equivalence serves as a first-order approximation of the aggregation process.

## C  Proof for Theorem 2

*Proof.* This proof first expands the model based on the feature and parameter matrices. Then, it identifies the representation updates of the topology and attributes within it. Finally, it establishes the relationship between these updated expressions and the cross-aggregation.

Firstly, the function $Softmax(\cdot)$ can be approximated using Random Features mappings, that is, $\mathbf{H} = Softmax(\mathbf{Q}\mathbf{K}^{\top})\mathbf{V} \approx \phi(\mathbf{Q})\phi(\mathbf{K})^{\top}\mathbf{V}$. Here, $\phi(\cdot)$ denotes a kernel-based feature mapping that linearizes the attention computation. In practice, the learnable projection matrix $\mathbf{W}$ applied to $\mathbf{X}$ can be viewed as a parametric approximation to this mapping.

By expanding node attributes $\mathbf{X} = [\mathbf{X}; \mathbf{P}]$, where $\mathbf{X} \in \mathbb{R}^{n \times f}$ and $\mathbf{P} \in \mathbb{R}^{n \times k}$, it can be obtained as

$$\mathbf{H} = \left( [\mathbf{X}; \mathbf{P}] \mathbf{W}^q (\mathbf{W}^k)^{\top} \left[ \begin{array}{c} \mathbf{X}^{\top} \\ \mathbf{P}^{\top} \end{array} \right] \right) [\mathbf{X}; \mathbf{P}] \mathbf{W}^v \tag{19}$$

Then, by expanding the Query (like $\mathbf{W}^q = \left[ \begin{array}{c} \mathbf{W}_1^q \\ \mathbf{W}_2^q \end{array} \right]$), and the Key and Value, it can be derived as

$$\begin{aligned} \mathbf{H} &= \left( (\mathbf{X}\mathbf{W}_1^q + \mathbf{P}\mathbf{W}_2^q) \left( (\mathbf{W}_1^k)^{\top}\mathbf{X}^{\top} + (\mathbf{W}_2^k)^{\top}\mathbf{P}^{\top} \right) \right) (\mathbf{X}\mathbf{W}_1^v + \mathbf{P}\mathbf{W}_2^v) \\ &= \left( (\mathbf{X}\mathbf{W}_1^q + \mathbf{P}\mathbf{W}_2^q) \left( (\mathbf{W}_1^k)^{\top}\mathbf{X}^{\top} + (\mathbf{W}_2^k)^{\top}\mathbf{P}^{\top} \right) \right) \mathbf{X}\mathbf{W}_1^v \\ &\quad + \left( (\mathbf{X}\mathbf{W}_1^q + \mathbf{P}\mathbf{W}_2^q) \left( (\mathbf{W}_1^k)^{\top}\mathbf{X}^{\top} + (\mathbf{W}_2^k)^{\top}\mathbf{P}^{\top} \right) \right) \mathbf{P}\mathbf{W}_2^v. \end{aligned} \tag{20}$$

It is evident that the equation includes several terms that describe the self-aggregation of topology and attributes. For instance, $\mathbf{P}\mathbf{W}_2^q(\mathbf{P}\mathbf{W}_2^k)^{\top}\mathbf{P}\mathbf{W}_2^v$ and $\mathbf{X}\mathbf{W}_1^q(\mathbf{X}\mathbf{W}_1^k)^{\top}\mathbf{X}\mathbf{W}_1^v$ stand for the self-aggregation of topology and attributes, respectively. Furthermore, this equation contains several terms that include cross-aggregation across topology and attributes. Exampled by the term $\mathbf{X}\mathbf{W}_1^q(\mathbf{P}\mathbf{W}_2^k)^{\top}\mathbf{P}\mathbf{W}_2^v$, by setting $\mathbf{Z} = \mathbf{X}\mathbf{W}_1^q$ and $\mathbf{B} = \mathbf{P}\mathbf{W}_2$, the cross-aggregation from the topology space to the attribute space can be obtained. Similarly, terms (*e.g.*, $\mathbf{P}\mathbf{W}_2^q(\mathbf{P}\mathbf{W}_2^k)^{\top}\mathbf{X}\mathbf{W}_1^v$) describing cross-aggregation from the attribute space to the topology space can be found. $\square$

## D  Proof for Theorem 3

*Proof.* This proof includes two main steps. First, we plan to derive the closed-form solutions for $\mathbf{Z}$ and $\mathbf{B}$ from the convex optimization objective (Eq. 15). These solutions are denoted as $\mathbf{Z}^*$ and $\mathbf{B}^*$, respectively. Second, we aim to establish the equivalence between these derived solutions $\mathbf{Z}^*$ and $\mathbf{B}^*$ and the feature updates in Eq. 10 and Eq. 11, respectively.

Let us denote the objective function (Eq. 15) as $\mathcal{O}(\mathbf{Z}, \mathbf{B})$, that is

$$\begin{aligned} \mathcal{O}(\mathbf{Z}, \mathbf{B}) &= \lambda \operatorname{Tr}(\mathbf{Z}^{\top}\tilde{\mathbf{L}}\mathbf{Z}) + \|\mathbf{B} - MLP(\mathbf{X})\|_F^2 - \eta\|\mathbf{Z}^{\top}\mathbf{B}\|_F^2 \\ &= \lambda \operatorname{Tr}(\mathbf{Z}^{\top}(\mathbf{I} - \tilde{\mathbf{A}})\mathbf{Z}) \\ &\quad + \operatorname{Tr}\left( (\mathbf{B} - MLP(\mathbf{X}))^{\top}(\mathbf{B} - MLP(\mathbf{X})) \right) \\ &\quad - \eta \operatorname{Tr}((\mathbf{Z}^{\top}\mathbf{B})^{\top}(\mathbf{Z}^{\top}\mathbf{B})) \end{aligned} \tag{21}$$

Firstly, the partial derivatives of $\mathcal{O}(\mathbf{Z}, \mathbf{B})$ with respect to $\mathbf{Z}$ can be calculated as

$$\frac{\partial \mathcal{O}(\mathbf{Z}, \mathbf{B})}{\partial \mathbf{Z}} = 2\lambda(\mathbf{I} - \tilde{\mathbf{A}})\mathbf{Z} - 2\eta(\mathbf{B}\mathbf{B}^{\top}\mathbf{Z}) \tag{22}$$

The closed-form solution $\mathbf{Z}^*$ for the objective function $\mathcal{O}$ can be derived by setting $\frac{\partial \mathcal{O}(\mathbf{Z},\mathbf{B})}{\partial \mathbf{Z}} = 0$, that is

$$2\lambda(\mathbf{I} - \tilde{\mathbf{A}})\mathbf{Z} - 2\eta(\mathbf{B}\mathbf{B}^\top\mathbf{Z}) = 0 \tag{23}$$

$$\Rightarrow \mathbf{Z}^* = \tilde{\mathbf{A}}\mathbf{Z} + \frac{\eta}{\lambda}\mathbf{B}\mathbf{B}^\top\mathbf{Z} \tag{24}$$

Then, by setting $\omega_1 = \frac{\eta}{\lambda+\eta}$, we obtain

$$\mathbf{Z}^* = (1 - \omega_1)\tilde{\mathbf{A}}\mathbf{Z} + \omega_1\mathbf{B}\mathbf{B}^\top\mathbf{Z} \tag{25}$$

For the update of topology representations in the proposed DCA module, *i.e.*, Eq. 10, the process can be rewritten as

$$\hat{\mathbf{Z}} = (1 - \tilde{\lambda})\tilde{\mathbf{A}}\mathbf{V} + \tilde{\lambda}\mathbf{S}\mathbf{V}, \tag{26}$$

where $\mathbf{S} \in \mathbb{R}^{n\times n}$ stands for the cross-attention score matrix, with the scores $s_{i,i} = \frac{n+\mathbf{q}_{i,:}\mathbf{k}_{i,:}^\top}{n+\sum_t \mathbf{q}_{i,:}\mathbf{k}_{t,:}^\top}$ and $s_{i,j} = \frac{\mathbf{q}_{i,:}\mathbf{k}_{j,:}^\top}{n+\sum_t \mathbf{q}_{i,:}\mathbf{k}_{t,:}^\top}$ for $i \neq j$.

To demonstrate the equivalence between Eq. 25 and Eq. 26, the first step is to set $\omega_1 = \tilde{\lambda}$. With this condition met, the required proof is to derive a matrix $\mathbf{B}$ that satisfies the equation $\mathbf{B}\mathbf{B}^\top = \mathbf{S}$.

Let us denote the diagonal matrix as $\mathbf{D}$, where $d_{i,i} = \sqrt{\frac{n+1}{n+\sum_t \mathbf{q}_{i,:}\mathbf{q}_{t,:}^\top}}$, the construction of $\mathbf{B} = \mathbf{D}\mathbf{Q}$ ensures that the above equation holds.

For the diagonal elements of $\mathbf{B}\mathbf{B}^\top$, there is

$$(\mathbf{B}\mathbf{B}^\top)_{i,i} = \sum_{j=0}^{d-1} b_{i,j}^2 = \sum_{j=0}^{d-1} (d_{i,i} \cdot q_{i,j})^2 = d_{i,i}^2 \sum_{j=0}^{d-1} q_{i,j}^2 \tag{27}$$

Given that the rows of $\mathbf{Q}$ are L2-normalized, we have $\sum_{j=0}^{d-1} q_{i,j}^2 = 1$. Due to the same source and parameter sharing, it follows that $\mathbf{Q} = \mathbf{K}$. Thus, we obtain $(\mathbf{B}\mathbf{B}^\top)_{i,i} = d_{i,i}^2 = \frac{n+1}{n+\sum_t \mathbf{q}_{i,:}\mathbf{q}_{t,:}^\top}$, which matches the definition of the score $s_{v,v}$ on the main diagonal.

For the off-diagonal elements of $\mathbf{B}\mathbf{B}^\top$ where $v \neq u$, there is

$$\begin{aligned}
(\mathbf{B}\mathbf{B}^\top)_{i,j} &= \sum_{t=0}^{d-1} b_{i,t} \cdot b_{j,t} = \sum_{t=0}^{d-1} (d_{i,i} \cdot q_{i,t})(d_{j,j} \cdot q_{j,t}) \\
&= d_{i,i} \cdot d_{j,j} \sum_{t=0}^{d-1} q_{i,t} \cdot q_{j,t} = c\frac{\mathbf{q}_{i,:}\mathbf{q}_{j,:}^\top}{n + \sum_k \mathbf{q}_{i,:}\mathbf{q}_{k,:}^\top}
\end{aligned} \tag{28}$$

Since $d_{i,i}$ and $d_{j,j}$ are the square roots of the denominators in the formula for $s_{i,i}$ and $s_{j,j}$, respectively, and $\mathbf{q}_{i,:}\mathbf{q}_{j,:}^\top$ is the dot product of the $i$-th and $j$-th rows of $\mathbf{Q}$, this matches the definition of $s_{i,j}$. Therefore, the correct construction of $\mathbf{B}$ should be $\mathbf{B} = \mathbf{D}\mathbf{Q}$.

Similarly, the closed-form solution $\mathbf{B}^*$ of the objective in Eq. 15 can be obtained by setting its derivative to 0 as

$$\frac{\partial \mathcal{O}(\mathbf{Z},\mathbf{B})}{\partial \mathbf{B}} = 2(\mathbf{B} - MLP(\mathbf{X})) + 2\eta(\mathbf{Z}\mathbf{Z}^\top\mathbf{B}) = 0 \tag{29}$$

$$\Rightarrow \mathbf{B}^* = MLP(\mathbf{X}) + \eta\mathbf{Z}\mathbf{Z}^\top\mathbf{B} \tag{30}$$

Then, by defining $\omega_2 = \frac{1}{1+\eta}$, there is

$$\mathbf{B}^* = (1 - \omega_2)MLP(\mathbf{X}) + \omega_2\mathbf{Z}\mathbf{Z}^\top\mathbf{B} \tag{31}$$

The proposed dual cross-attention module for updating the attribute representations, *i.e.*, Eq. 11, can be rephrased as

$$\hat{\mathbf{B}} = (1 - \tilde{\gamma})MLP(\mathbf{X}) + \tilde{\gamma}\mathbf{S}\mathbf{V} \tag{32}$$

Similarly, considering that matrix $\mathbf{S}$ maintains the same structure as described in Eq. 26, the crucial step to ensure $\mathbf{B}^* = \hat{\mathbf{B}}$ is to establish the parameter $\omega_2 = \tilde{\gamma}$ and to set $\mathbf{Z} = \mathbf{D}\mathbf{Q}$. This approach guarantees that the necessary conditions for the equivalence are met. $\square$

# E    Experimental Details

## E.1    Datasets and Splitting

**Datasets.** In the node classification experiments, sixteen publicly available benchmark datasets are utilized. These graphs can be classified into two categories based on whether their Edge Homophily [37] exceeds 0.5: seven graphs are tagged as *homophilic graphs*, including Cora [42], CiteSeer [42], PubMed [42], Photo [43], CS [43], Physics [43], and Questions [38]. The remaining seven graphs are marked as *heterophilic graphs*, containing Cornell [37], Texas [37], Wisconsin [37], Actor [45], Chameleon [40], Squirrel [40], and Ratings [38]. It is worth noting that the original Chameleon and Squirrel exhibit neighborhood overlap, and are thus filtered according to the study [38]. Additionally, two large-scale graph datasets, *i.e.*, ogbn-arxiv [23] and ogbn-proteins [23], are employed for node property prediction experiment. Statistics are shown in Tab. 4.

Table 4: Statistics of sixteen graph datasets. $\#h$ denotes the edge homophily shown in [37].

| Dataset | Nodes | Edges | Features | Classes | $\#h$ |
|---|---|---|---|---|---|
| Cora | 2,708 | 5,278 | 1,433 | 7 | 0.81 |
| CiteSeer | 3,327 | 4,552 | 3,703 | 6 | 0.74 |
| PubMed | 19,717 | 44,324 | 500 | 3 | 0.80 |
| Photo | 7,650 | 238,163 | 745 | 8 | 0.83 |
| CS | 18,333 | 81,894 | 6,805 | 15 | 0.81 |
| Physics | 34,493 | 247,962 | 8,415 | 5 | 0.93 |
| Questions | 48,921 | 153,540 | 301 | 2 | 0.84 |
| Cornell | 183 | 280 | 1,703 | 5 | 0.30 |
| Texas | 183 | 295 | 1,703 | 5 | 0.11 |
| Wisconsin | 251 | 466 | 1,703 | 5 | 0.21 |
| Actor | 7,600 | 33,544 | 931 | 5 | 0.22 |
| Chameleon | 890 | 8,854 | 2,325 | 5 | 0.24 |
| Squirrel | 2,223 | 46,998 | 2,089 | 5 | 0.21 |
| Ratings | 24,492 | 93,050 | 300 | 5 | 0.38 |
| ogbn-proteins | 132,534 | 39,561,252 | 8 | 2 | 0.38 |
| ogbn-arxiv | 169,343 | 1,157,799 | 300 | 128 | 0.65 |

**Dataset Splitting.** To ensure that the experimental results are credible and reproducible, this paper follows well-established dataset splitting strategies. For the Cora, CiteSeer, and PubMed, the public standard splitting described in [29] is adopted, with 20 nodes per class for training, 500 for validation, and 1000 for testing. The Photo, CS, and Physics are randomly divided into training, validation, and testing sets in a 60%, 20%, and 20% ratio, respectively. For the heterophilic datasets Cornell, Texas, Wisconsin, Actor, and Chameleon, this paper employ 10 standard train/validation/test splits with a division ratio of 48%, 32%, and 20%, respectively. Note that the Chameleon and Squirrel used here are duplicates-removed filtered versions as referenced in [38]. The Ratings, and Questions follow a 50%/25%/25% train/validation/test random split pattern. For the two datasets from the OGB [23], *i.e.*, ogbn-arxiv and ogbn-proteins, the provided standard splits are utilized.

## E.2    Introduction of Baselines

The comparative analysis in the experiments involves seven Graph Neural Networks (GNNs) as well as seven Graph Transformers (GTs) as the baseline models. To be specific, the GNNs include four *standard GNNs*, *i.e.*, GCN [29], GAT [47], GraphSAGE [20], and APPNP [30], and two *universal GNNs* for graphs with diverse homophily, *i.e.*, GPR-GNN [6] and GloGNN [33], and a *non-message-passing GNN* with a separate topology and attribute design, *i.e.*, LINKX [34]. Besides, the baseline GTs encompass eight state-of-the art models, namely, GraphGPS [39], NAGphormer [2], Exphormer [44], GOAT [31], NodeFormer [49], SGFormer [50], Polynormer [7], and Gradformer [36]. These models are implemented following the released code of the original paper.

### E.2.1    Graph Neural Networks (GNNs)

The following specifies the GNN baselines employed in our comparative analysis.

- GCN [29]: A seminal GNN that integrates graph topology and node attributes via graph convolution.
- GAT [47]: A classic graph attention network that weights propagation using an attention mechanism.
- GraphSAGE [20]: A scalable variant of GCN that employs neighbor sampling and diverse aggregation strategies.
- APPNP [30]: A variant of GCN that weights propagation based on personalized PageRank.
- GPR-GNN [6]: A universal variant of GCN that weights propagation using learnable layer coefficients.
- LINKX [34]: A non-message-passing GNN that directly combines representations of graph topology and node attributes.
- GloGNN [33]: A universal variant of GCN that obtains the propagation matrix from an optimization objective describing node relationships.

### E.2.2 Graph Transformers (GTs)

The following specifies the GT baselines utilized in our comparative analysis.

- GraphGPS [39]: A general GT architecture incorporating positional encodings and local/global modules.
- NodeFormer [49]: A scalable GT architecture that learns layer-specific graph structures via a kernelized Gumbel-Softmax operator.
- NAGphormer [2]: A creative GT architecture constructing token vectors using neighborhood aggregation.
- Exphormer [44]: A general GT architecture combining local, extended, and virtual node-based global attention.
- GOAT [31]: A comprehensive GT architecture linearizing computational complexity based on the k-means algorithm.
- SGFormer [50]: A lightweight GT architecture featuring a single-layer self-attention module.
- Polynormer [7]: A polynomial-expressive GT architecture learning high-degree polynomials on input features.
- Gradformer [36]: An effective GT architecture dynamically modeling node relationships by exponentially diminishing values in the decay mask matrix.

For the four GNN baselines, including GCN, GAT, GraphSAGE, and APPNP, we utilize the public library, PyTorch Geometric (PyG) [13], for their implementation. For the other three GNN baselines, we utilize their original code. The sources are outlined as

- GPR-GNN: https://github.com/jianhao2016/GPRGNN
- LINKX: https://github.com/CUAI/Non-Homophily-Large-Scale
- GloGNN: https://github.com/RecklessRonan/GloGNN

For the GT baselines, including GraphGPS, NodeFormer, NAGphormer, Exphormer, GOAT, SGFormer, Polynormer, and Gradformer, we utilize their source code. The sources are detailed as

- GraphGPS: https://github.com/rampasek/GraphGPS
- NodeFormer: https://github.com/qitianwu/NodeFormer
- NAGphormer: https://github.com/JHL-HUST/NAGphormer
- Exphormer: https://github.com/hamed1375/Exphormer
- GOAT: https://github.com/devnkong/GOAT
- SGFormer: https://github.com/qitianwu/SGFormer
- Polynormer: https://github.com/cornell-zhang/Polynormer
- Gradformer: https://github.com/LiuChuang0059/Gradformer

### E.3 Experimental Setups

**Configurations.** The experiment is performed on two Linux machines using a single GeForce RTX4090 24 GB GPU and a single NVIDIA A800 80GB GPU, respectively. The reported results are averaged over ten random trials. All models operate under a semi-supervised learning paradigm, where the results on validation sets are referenced to fine-tune hyperparameters.

Table 5: Hyperparameters of UGCFormer per dataset.

| Dataset | # layers $l$ | # dimensions $d$ | lr | $\alpha$ | $\beta$ | wd |
|---|---|---|---|---|---|---|
| Cora | 4 | 256 | 0.001 | 0.5 | 1 | 5e-3 |
| CiteSeer | 3 | 256 | 0.001 | 0.6 | 0.1 | 5e-3 |
| PubMed | 3 | 128 | 0.001 | 0.5 | 0.1 | 1e-2 |
| Photo | 4 | 512 | 0.001 | 0.3 | 0.1 | 5e-5 |
| CS | 3 | 512 | 0.001 | 0.6 | 0.1 | 5e-4 |
| Physics | 3 | 512 | 0.001 | 0.5 | 1 | 5e-4 |
| Questions | 2 | 256 | 0.005 | 0.3 | 0.1 | 5e-4 |
| Cornell | 5 | 256 | 0.001 | 0.9 | 0.001 | 1e-2 |
| Texas | 4 | 128 | 0.001 | 0.7 | 0.001 | 1e-2 |
| Wisconsin | 4 | 128 | 0.001 | 0.9 | 0.001 | 1e-2 |
| Actor | 5 | 512 | 0.001 | 0.9 | 0.01 | 1e-2 |
| Chameleon | 2 | 512 | 0.001 | 0.2 | 0.01 | 5e-3 |
| Squirrel | 4 | 512 | 0.001 | 0.2 | 0.01 | 5e-5 |
| Ratings | 2 | 64 | 0.001 | 0.3 | 0.001 | 0 |
| ogbn-proteins | 1 | 128 | 0.001 | 0.7 | 0.01 | 5e-4 |
| ogbn-arxiv | 2 | 512 | 0.01 | 0.3 | 0.01 | 5e-4 |

**Hyper-parameters.** The hyperparameters are selected via a grid search strategy. In the node classification task, models are trained employing an Adam optimizer with the learning rate among $\{0.001, 0.005, 0.01\}$ and the weight decay among $\{0, 1e-5, 5e-5, 1e-4, 5e-4, 1e-3, 5e-3, 1e-2\}$. The number of layers is selected from $\{1, 2, 3, 4, 5\}$, and the dimension of hidden layers is chosen from $\{64, 128, 256, 512\}$, and their impacts on model performance are analyzed in Section 4.2. For the node property prediction task, the hyperparameter selection follows the baseline [50]. For the unique hyperparameters in UGCFormer, $\alpha$ is chosen from a range starting at $0.1$ and increasing by increments of $0.1$, up to $0.9$, $\beta$ is selected from $\{0.001, 0.01, 0.1, 1\}$, and $\tau$ is fixed to $0.5$. Refer to Tab. 5 for the chosen parameters that correspond to the reported results.

Table 6: Training time and GPU memory usage on three graphs.

| Method | CiteSeer | | PubMed | | ogbn-arxiv | |
|---|---|---|---|---|---|---|
| | Train/Epoch (ms) | Mem. (MB) | Train/Epoch (ms) | Mem. (MB) | Train/Epoch (ms) | Mem. (MB) |
| GraphGPS | 16.82 | 140 | 46.99 | 470 | 166.10 | 8,102 |
| NodeFormer | 10.20 | 110 | 11.14 | 218 | 84.00 | 2,066 |
| NAGphormer | 10.81 | 166 | 17.65 | 352 | 760.50 | 1,962 |
| Exphormer | 14.20 | 159 | 25.00 | 696 | 145.70 | 6,758 |
| SGFormer | 5.80 | 84 | 6.07 | 142 | 36.90 | 1,024 |
| Polynormer | 9.20 | 174 | 15.20 | 307 | 170.07 | 4,729 |
| UGCFormer | 8.80 | 106 | 10.60 | 246 | 69.60 | 2,050 |

### E.4 Additional experiment results

**Running Time and Space Consumption**. To further illustrate the efficiency and scalability of the proposed UGCFormer, this experiment compares it with other GTs in terms of runtime and GPU memory usage. Common hyperparameters are uniformly applied across all models to highlight the impact of their components, particularly the attention modules. As depicted in Table 6, UGCFormer consistently has the second-lowest running time and ranks among the top three in terms of lowest GPU memory usage across three datasets. Despite utilizing linearized attention mechanisms, most linearized GTs, including GraphGPS, NodeFormer, Exphormer, and Polynormer, perform worse than UGCFormer. This highlights the lightweight and efficient design of UGCFormer.

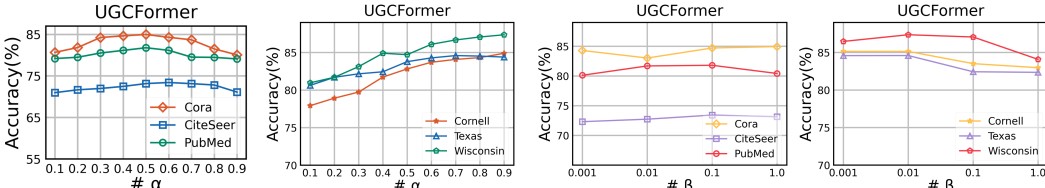

Figure 6: Performance variations for varying $\alpha$.     Figure 7: Performance variations for varying $\beta$.

**Hyperparameter $\alpha$.** As depicted in the Fig. 6, UGCFormer exhibits stable performance within a specific parameter range for each dataset. For instance, on Cora, performance variation is minimal within the set $\{0.3, 0.4, 0.5, 0.6\}$, indicating that the model is robust to changes in hyperparameter $\alpha$. Similar observations are made for heterophilic graphs.

**Hyperparameter $\beta$.** From the Fig. 7, it can be observed that the performance remains stable across the selection range of $\beta$, demonstrating that the model is not sensitive to this parameter. Even in the worst case of parameter selection, the model achieves performance that is comparable to the baseline.

## F   Discussion

**Comparison with Edge-augmented Graph Transformer (EGT).** The proposed UGCFormer shares a conceptual connection with EGT [27], yet their core mechanisms differ substantially. EGT augments pairwise attribute-based attention between nodes using graph topology, whereas UGCFormer captures the interaction between topology and attributes through a cross-attention mechanism. Although EGT introduces an additional edge channel alongside the attribute channel, this design primarily aims to utilize edge features to modulate the attention process (via addition or gating) rather than to explicitly model the interaction between the two channels. Moreover, EGT requires large-scale edge features of size $O(n^2 d)$ (where $n$ denotes the number of nodes and $d$ the feature dimension), which significantly increases computational complexity and limits scalability. In contrast, UGCFormer adopts a linear cross-attention module that efficiently models topology-attribute interactions with linear time and space complexity.

**Broader Relation to Other Categories of Graph Transformers.** The proposed cross-aggregation mechanism can also provide a unified interpretation for other two types of Graph Transformers, that is, *those based on context-node sampling or edge rewriting*. Although both categories of GTs are implemented in different ways, their essence is to determine the subgraph for each node to perform local message passing (via graph convolution or self-attention). Thus, they can be uniformly expressed by obtaining the graph structure. As a result, the topology representation can be straightforwardly determined based on the eigenvalue decomposition of the adjacency matrix, as described in the manuscript. Ultimately, a formulation similar to that of cross-aggregation can be derived. Therefore, their underlying mechanism can be attributed to a cross-aggregation between topology and attribute representations.

## G   Limitations

The proposed UGCFormer, like other Graph Transformers (GTs), relies on a fixed set of hyperparameters, requiring significant tuning and prior knowledge to optimize results. This not only limits their applicability but also increases computational costs. Future work should explore adaptive learning mechanisms to automate hyperparameter adjustment based on input data characteristics, reducing manual intervention and enhancing generalizability.

