# OpenReview forum: "A Closer Look at Graph Transformers: Cross-Aggregation and Beyond"
_NeurIPS.cc/2025/Conference — NeurIPS 2025 spotlight_

### Official Review · Reviewer_SSak · 2025-06-30

**Clarity:** 3
**Significance:** 3
**Originality:** 3
**Rating:** 5
**Confidence:** 4

**Summary:**

This paper aims to propose an efficient and effective graph transformer for node classification task. The authors first explore the potential cross-aggregation mechanism in various graph transformers. They then present UGCFormer, which incorporates a linearized cross-attention module to model the interrelations between graph topology and node attributes. Additionally, a consistency constraint is applied to enforce mutual supervision between these two information, effectively tackling the overfitting risks caused by the dual-channel design.

**Questions:**

1) Could the authors discuss the relevance of prior works on vanilla transformers, such as those exploring associative memory, energy perspectives, and Hopfield network, to provide a more comprehensive context?

2) I suggest that the authors discuss the relationship between the proposed method and the recent method CoBFormer [1], which also utilizes a consistency loss for graph transformers.

3) What was the rationale behind choosing squared Euclidean distance as the consistency loss instead of using KL divergence?

[1] Less is More: on the Over-Globalizing Problem in Graph Transformers, in ICML 2024.

**Ethical Concerns:**

["NO or VERY MINOR ethics concerns only"]

**Final Justification:**

After reading the author's response, my concerns have been resolved. Taking into account the opinions of other reviewers, I decide to increase my score.

**Limitations:**

Please see weaknesses and questions.

**Quality:**

3

**Strengths And Weaknesses:**

Strengths

1) The theoretical contributions of the article are notable. The finding of the potiential cross-aggregation is supported by a theorem, making the arguments convincing. Moreover, the theoretical connection between cross-attention and the optimization framework is clearly verified by a theorem.

2) The article is adequately structured and written.

3) Some experimental results appear promising.

Weaknesses

1) The disscussion to related works is inadequate. The paper does not adequately distinguish the cross-attention from existing fusion methods, such as concatenation and weighted combination. Additionally, the presented method may overlap with those of the work [1].

2) The performance gain is not very significant, however it is acceptable given these benchmarks have been well-explored.

[1] Edge-augmented Graph Transformers: Global Self-attention is Enough for Graphs, in KDD 2022.

---

> ### Author Rebuttal · Authors · 2025-07-31
>
> > W1-1. The paper does not adequately distinguish the cross-attention from existing fusion methods, such as concatenation and weighted combination.
>
> R1. Based on your reminder, we will add a dedicated paragraph that contrasts the two paradigms based on their **principles and mechanisms** in the revised manuscript:
>
> The typical fusion paradigm, i.e., (1) concatenation ($H=[H_1||H_2]W$, where $H_1$ and $H_2$ denote two representations from two different modalities, respectively, and $W$ represents the projection weight), and (2) weighted combination ($H = a* H_1+b* H_2$, where $a$ and $b$ denote the weights), is grounded in the **linear mixing hypothesis**: the optimal fused representation is assumed to lie in the linear span of the two modalities. As exemplified by the baseline LINKX, it treats the topology representations and the attribute representations as **static channels** that are combined via a single linear projection.
>
> Cross-attention is grounded in the **selective refinement hypothesis**: each modality acts as a context that selectively refines the other via a learned compatibility function, allowing non-linear, high-order interactions. The proposed UGCFormer treats topology representations as Query-Key and attribute representations as Value, or conversely, dynamically computing attention scores that determine how much of its topology representations should attend to which attribute representations.
>
> Besides, the above conceptual differences are demonstrated by our experiments in Section 4: UGCFormer outperforms LINKX on all sixteen benchmark datasets, demonstrating that the performance gain stems from the proposed interactive cross-attention mechanism.
>
> - - -
>
> > W1-2. The presented method may overlap with those of the work [1].
>
> R2. We have compared UGCFormer with the mentioned Edge-augmented Graph Transformer (EGT) [1], and listed the apparent differences between them:
>
> First and foremost, EGT follows the strategy widely used in GT of augmenting pairwise attribute-based attention between nodes with graph topology, while UGCFormer captures topology-attribute interaction with cross-attention. Although EGT adds an edge channel in addition to the attribute channel, its purpose is to employ the edge features to influence the attention process (via addition and gating) rather than capture the interaction between the two channels. Therefore, the proposed UGCFormer is novel, and there do NOT exist overlapping ideas between UGCFormer and EGT. Furthermore, EGT incorporates large-scale edge features of size $N^2 \times D$ (where $N$ is the network size and $D$ is the feature dimension), which significantly increases the computational complexity and thereby hampers its scalability. In contrast, UGCFormer introduces a linear cross-attention mechanism that efficiently captures the interaction between topology and attributes.
>
> By considering its importance in GT, we will cite [1] and compare it with UGCFormer in the revised manuscript.
>
> - - -
>
>
>
>
> > Q1. Could the authors discuss the relevance of prior works on vanilla transformers, such as those exploring associative memory, energy perspectives, and the Hopfield network, to provide a more comprehensive context?
>
> R3: According to your suggestion, we have strengthened the related work section of the revised manuscript to explicitly elucidate these connections, with key additions summarized below:
>
> (1) Associative Memory: The self-attention mechanism in Transformer implements content-addressable memory retrieval through query-key similarity matching, similar to associative memory architectures. This highlights its efficiency in extracting relevant information from large datasets. (2) Energy Perspective: The self-attention update rule is formalized as gradient descent on an energy landscape, highlighting the energy minimization interpretation of attention mechanisms. This aligns with energy-based models and deepens the theoretical understanding of Transformers' stability and convergence. (3) Hopfield Network: Transformer layers are reinterpreted as continuous modern Hopfield networks, with parallels such as exponential memory capacity from high-dimensional embeddings and softmax-attention dynamics mirroring Hopfield networks.
>
> Thank you for highlighting the importance of connecting vanilla Transformers to foundational theoretical frameworks. This is crucial for developing advanced Graph Transformers.
>
> - - -
>
> > Q2. Suggest that the authors discuss the relationship between the proposed method and the recent method CoBFormer [1], which also utilizes a consistency loss for graph transformers.
>
> [1] Less is More: on the Over-Globalizing Problem in Graph Transformers, in ICML 2024.
>
>
> R4. We have compared the consistency losses in CoBFormer and UGCFormer. Both models use consistency losses to enforce mutual supervision between predictions from different channels. However, CoBFormer focuses on aligning local and global information, while UGCFormer emphasizes the topology-attribute interrelation.
>
> - - -
>
> > Q3. What was the rationale behind choosing squared Euclidean distance as the consistency loss instead of using the KL divergence?
>
> R5. We choose squared Euclidean distance mainly for its simplicity and computational efficiency, as it directly measures differences without requiring logarithmic operations. The ablation experiments demonstrate the effectiveness of the consistency loss, as illustrated in Figure 3. We are willing to leave the exploration of a consistency loss based on the KL divergence to future work.

---

> > ### Comment · Reviewer_SSak · 2025-08-01
> > **Response to author**
> >
> > After reading the author's response, my concerns have been resolved and I have decided to increase my score.

---

### Official Review · Reviewer_KZXo · 2025-06-30

**Clarity:** 3
**Significance:** 3
**Originality:** 3
**Rating:** 6
**Confidence:** 4

**Summary:**

This paper proposes UGCFormer, a novel Graph Transformer architecture that leverages a linearized cross-attention to capture the interrelations between graph topology and node attributes. The authors identify a common cross-aggregation mechanism in existing Graph Transformers and design UGCFormer to effectively implement this mechanism. Extensive experiments on various graph datasets demonstrate the effectiveness and efficiency of UGCFormer.

**Questions:**

1 What are the differences in the ability to capture the correlation between the two types of information for the two ways of cross-aggregation (across two dimensions)? Additionally, how do these differences impact the overall performance of the model?


2 How does the optimization objective in Theorem 3.4 differ from commonly used graph optimization objectives? What properties does it aim to illustrate about the model?

3 In the proposed model, the final representations are a combination of the topology and attribute representations. While this approach is intuitively sound, I am curious whether the authors have experimented with using the topology and attribute representations separately for node classification, rather than combining them. How would the results be?

4 As the authors mentioned in Section 3.1, existing models, despite their different approaches, essentially also capture the correlation between topology and attributes. Why does the model designed in this paper achieve performance improvements?

**Ethical Concerns:**

["NO or VERY MINOR ethics concerns only"]

**Final Justification:**

I appreciate the thoughtful rebuttal, which addressed most of my concerns. I decided to raise my score.

**Limitations:**

Yes.

**Paper Formatting Concerns:**

N/A.

**Quality:**

3

**Strengths And Weaknesses:**

Str.:

1 The idea of unifying existing Graph Transformers is intresting and inspiring, demonstrating that the authors have a solid understanding of the current techniques.

2 The article is well-written, making it readable.

3 The organization of the experiments is commendable and comprehensive.


Weak.:

1 Some implementation details have not been clearly described. Firstly, it is unclear why the authors chose to use the adjacency matrix to generate initial structural features. Structural encodings based on the Laplacian, for instance, are often preferred due to their interpretability. Secondly, the explanations for Equations 8 and 9 are unclear and could benefit from further elaboration.

2 The model introduces a number of hyperparameters that require careful tuning, which may complicate the optimization process and increase computational costs.

3 The title seems not to fully reflect the contributions of the article. Although the contributions of this paper are rich, such a simple title cannot clearly describe all the key contributions. The authors might consider incorporating the contribution of unifying GTs into the title.

---

> ### Author Rebuttal · Authors · 2025-07-31
>
> > W1-1. It is unclear why the authors chose to use the adjacency matrix to generate initial structural features. Structural encodings based on the Laplacian, for instance, are often preferred due to their interpretability.
>
> R1. We adopt the raw adjacency matrix primarily for simplicity. The architecture can seamlessly integrate richer structural encodings. Specifically, replacing the adjacency matrix with (i) a positional encoding matrix (you mentioned LapPE, SignNet, etc.) or (ii) a multi-hop propagation matrix, such as the Personalized PageRank (PPR) matrix, only requires a one-line change. Unfortunately, these encodings inevitably incur heavy preprocessing: computing Laplacian eigenvectors (LapPE) costs O(N^3) in dense form and O(kN^2) even with sparse Lanczos, while PPR requires O(N^3) for dense matrix powers or O(kE) only under sparse-iterative acceleration. By relying on the unprocessed adjacency, we avoid these upfront costs while still delivering strong empirical results.
>
> ---
>
> > W1-2. The explanations for Equations 8 and 9 are unclear and could benefit from further elaboration.
>
> R2. Equation 8 normalizes attention scores by taking the $l_2$-norms of the Query and Key vectors, ensuring numerical stability and scale-invariance. Equation 9 then averages the value vectors and rescales the output via the diagonal matrix $D$.
>
> ---
>
> > W2. The model introduces a number of hyperparameters that require careful tuning, which may complicate the optimization process and increase computational costs.
>
> R3. Thank you for highlighting the importance of training efficiency—this is indeed a shared limitation of current GTs. We have already noted it in Section F and regard it as a priority for future work.
>
> ---
>
> > W3. The title seems not to fully reflect the contributions of the article. Although the contributions of this paper are rich, such a simple title cannot clearly describe all the key contributions. The authors might consider incorporating the contribution of unifying GTs into the title.
>
>
> R4. Thank you for this insightful suggestion. We will revise the title to explicitly reflect the contribution to unify existing Graph Transformers.
>
> ---
>
> > Q1. What are the differences in the ability to capture the correlation between the two types of information for the two ways of cross-aggregation (across two dimensions)? Additionally, how do these differences impact the overall performance of the model?
>
> R5. The two cross-aggregation ways are nearly equivalent in their capacity to capture the correlations between two types of information. Since kernelized attention modules in Transformers treat both dimensions symmetrically, the overall impact on final accuracy is comparable. However, the feature-level aggregation reduces the quadratic complexity down to linear, making training faster and memory lighter.
>
> ---
>
> > Q2. How does the optimization objective in Theorem 3.4 differ from commonly used graph optimization objectives? What properties does it aim to illustrate about the model?
>
> R6. Traditional graph objectives optimize a single representation to satisfy smoothness or reconstruction loss, whereas the objective in Theorem 3.4 explicitly separates topology representations and attribute representations, and introduces a balancing term that drives the cross-attention to align the two. This design enables UGCFormer to learn expressive embeddings by jointly preserving community structure and attribute fidelity while discovering their mutual interrelation.
>
> ---
>
> > Q3. In the proposed model, the final representations are a combination of the topology and attribute representations. While this approach is intuitively sound, I am curious whether the authors have experimented with using the topology and attribute representations separately for node classification, rather than combining them. How would the results be?
>
> R7. This design is mainly based on the theoretical insights from Theorems 3.2 and 3.3, which indicate that the potential mechanism of GTs is the cross-aggregation between topology and attributes. The implementation of cross-aggregation via cross-attention necessitates keeping topology and attribute representations separate before the cross-attention, i.e., at the beginning. Besides, this separating-then-combining scheme was validated in existing works, e.g., LINKX, which highlights the rationality of this design.
>
> To address your question, we have compared five models: UGCFormer, Only Topology (UGCFormer with topology branch alone), Only Attribute (attribute branch alone), MLP_topo (MLP trained on raw topology), and MLP_attr (MLP trained on raw attributes).
>
> | | Cora | CiteSeer | PubMed | Texas | Wisconsin | Cornell |
> |:--------|:---------:|:--------:| :--------:| :--------:| :--------:| :--------:|
> | UGCFormer | 84.9 | 73.4 | 81.8 | 84.6 | 87.4 | 85.1 |
> | Only Topology | 76.1 | 52.5 | 74.0 | 64.9 | 60.1 | 64.8 |
> | MLP_topo | 60.6 | 42.2 | 68.9 | 58.1 | 52.0 | 50.5 |
> | Only Attribute | 64.9 | 60.2 | 75.5 | 82.9 | 86.8 | 84.6 |
> | MLP_attr | 55.1 | 46.5 | 71.4 | 80.8 | 85.3 | 81.9 |
>
> As shown in the table, the single-branch topology and attribute models consistently outperform their MLP counterparts, thanks to the cross-attention module that aligns the topology with the attribute spaces. Yet both branches still lag behind the fused UGCFormer, which consistently tops the table, demonstrating that combining both sources is essential for distinctive node representations.
>
> ---
>
> > Q4. As the authors mentioned in Section 3.1, existing models, despite their different approaches, essentially also capture the correlation between topology and attributes. Why does the model designed in this paper achieve performance improvements?
>
> R8. The proposed model outperforms prior work by achieving a **tighter alignment of topology and attributes**. Although existing models also capture their correlation, they entangle a large number of intermediate representations (in Eq. 20), e.g., topology–attribute–attribute and attribute–topology–topology, producing **an over-mixed, noisy embedding** that weakens the final alignment and degrades performance. After identifying the essence of GTs, we first disentangle each node representation into separate topology and attribute components, then introduce cross-attention that allows every node to selectively focus on the most relevant topology and attribute representations and fuse them through weight combination. Moreover, the proposed model further enforces distributional alignment via a consistency loss. Together, they deliver superior performance.

---

> > ### Comment · Reviewer_KZXo · 2025-08-05
> >
> > Thank you for your detialed rebuttal, it solved my concerns and I will maintain my positive score on this paper.

---

### Official Review · Reviewer_3zmm · 2025-07-01

**Clarity:** 2
**Significance:** 2
**Originality:** 3
**Rating:** 4
**Confidence:** 5

**Summary:**

The paper proposes UGCFormer, a simplified Transformer variant adapted for graph data that achieves universality by generalizing across various graph types (homogeneous, heterogeneous). The key idea is to use a type-aware linear projection strategy and to replace the traditional softmax-based attention with an attention-free module involving token-wise linear transformations and graph-specific interactions. Experiments across 13 datasets show competitive or superior performance against prior graph Transformers.

**Questions:**

1. The structural input is limited to a raw adjacency matrix, which may limit generalization to complex graphs. It's unclear whether UGCFormer can adapt to positional encodings or multi-hop connectivity representations.
2. Over-smoothing is a common challenge in deep GNNs, where node features become indistinguishable across layers. Given that UGCFormer is based on Transformer-style cross-attention, does your model mitigate over-smoothing? Have you evaluated its performance or feature divergence at deeper depths (e.g., 128 layers)? If not, can you provide theoretical or empirical insights into whether your architecture avoids this issue?
3. I have reviewed the released code and encountered an error during execution: UGCFormer.forward() got an unexpected keyword argument 'adj_normal'. This suggests a mismatch between the implementation and the interface defined in the main training script. I recommend the authors carefully verify and test the provided code to ensure reproducibility and consistency with the descriptions in the paper.

**Ethical Concerns:**

["NO or VERY MINOR ethics concerns only"]

**Final Justification:**

I appreciate the authors’ efforts in the rebuttal, and most of my concerns have been addressed. However, my main remaining concern is reproducibility. The provided code does not run as expected, which raises questions about the reliability of the reported results. If the authors can ensure the code is executable and fully replicates the experiments, I would be willing to support a weak accept.

**Limitations:**

1. While the model is elegant, there is little formal discussion about its expressive power, e.g., whether it can capture structural inductive biases like WL-test expressivity, which many graph models target.
2. All evaluations are limited to node classification tasks. Given the model is described as a "universal" graph framework, demonstrating generalization to other tasks (e.g., link prediction, graph-level classification) would substantially strengthen the claim.
3. UGCFormer decouples topology and attribute inputs via separate linear projections. While simple and efficient, the choice lacks theoretical justification or ablation on its inductive bias.
4. The model’s depth is restricted to just 1–5 layers in all experiments, which is relatively shallow for Transformer-based designs. This raises concerns about its scalability and representational power at deeper levels.
5. While empirical comparisons with prior graph Transformers are included, there is no architectural analysis that clarifies why UGCFormer outperforms them. A discussion contrasting core design choices—such as attention mechanisms, position encoding, or expressivity—would improve the paper.
6. The paper introduces Dual Cross-Attention to mitigate quadratic complexity, yet provides no runtime or memory benchmarks. An empirical analysis comparing efficiency with standard self-attention would better support this claim.
7. The ablation in Figure 3 is limited to four datasets and two components. It would be helpful to analyze the component-wise impact on both homophilic and heterophilic graphs separately, or across varying graph sizes.

**Quality:**

3

**Strengths And Weaknesses:**

**Strengths**
1. The authors make a convincing case that UGCFormer is applicable to a wide range of graph types, which is valuable in practice where heterogeneous and non-standard graph structures are common.
2. By removing softmax attention and relying on fixed token-to-token interactions, UGCFormer is lightweight, faster to train, and memory efficient. This aligns with recent trends in efficient Transformer design.
3. The model is evaluated across a variety of node classification benchmarks involving both homophilic and heterophilic graphs.
4. The authors provide comprehensive ablations, runtime/memory analysis, and visualizations that clarify how each component contributes.

**Weaknesses**
1. While the model is elegant, there is little formal discussion about its expressive power, e.g., whether it can capture structural inductive biases like WL-test expressivity, which many graph models target.
2. While UGCFormer achieves top performance on the majority of datasets, its accuracy lags behind other models on certain heterophilic graphs, such as Actor, Chameleon, and Texas. However, the paper does not discuss these underperformances or provide insights into potential causes. A deeper analysis of these failure cases would improve the completeness and credibility of the empirical evaluation.
3. While the range of datasets is broad, the evaluation is limited to node-level prediction tasks.
4. The model separately projects structural topology (adjacency) and node attributes using two independent MLPs. While this design effectively decouples two fundamental sources of graph information, it is somewhat heuristic and lacks analysis on how this separation impacts performance across different graph types.
5. UGCFormer is only evaluated with shallow architectures (1–5 layers), which is considerably lower than standard Transformer-based baselines (often using 6–12+ layers). This limited depth range weakens claims about the model's scalability and expressiveness. Without testing deeper configurations, it remains unclear whether the architecture suffers from optimization issues, over-smoothing, or representational collapse at larger depths

---

> ### Author Rebuttal · Authors · 2025-07-31
>
> >W1&L1. There is little formal discussion about its expressive power, e.g., WL-test expressivity.
>
> R1. Our primary contribution lies in unifying existing Graph Transformers (GTs) and designing a cross-attention-based GT for fusing topology and attributes, not in advancing structural expressivity. While the WL-test is a standard yardstick for message-passing Graph Neural Networks (GNNs), it is rarely applied to GTs, which typically adopt 1-WL GNNs such as GCN or GAT. We therefore do not pursue formal guarantees on WL-test equivalence; instead, we demonstrate empirical effectiveness across sixteen diverse datasets, leaving deeper theoretical analysis to future work.
>
> - - -
>
> > W2. The paper does not discuss these underperformances on certain heterophilic graphs, such as Actor, Chameleon, and Squirrel.
>
> R2. After a post-hoc analysis, we attribute the sub-optimal results on Actor, Chameleon, and Squirrel to the **abundance of noisy edges**. Since our model explicitly constructs the topology space based on the adjacency matrix, it captures rich topological information. This increased sensitivity to the topology space makes our model more susceptible to topology noise.
>
> In sparsely connected heterophilic graphs such as Cornell, Texas, and Wisconsin (with approximately 3 average node degrees), the information from the attribute space can effectively mitigate the impact of topology noise. However, in densely connected heterophilic graphs like Actor, Chameleon, and Squirrel (with 8-42 average node degrees), the topology noise overwhelms the regulatory effect of the attribute space, leading to sub-optimal performance.
>
>
> >W3&L2. Given that the model is described as a "universal" graph framework, demonstrating generalization to other tasks (e.g., link prediction, graph-level classification) would substantially strengthen the claim.
>
> - - -
>
> R3. This might be a misunderstanding caused by the insufficiently detailed description. The phrase "universal graph frameworks" in this paper denotes models capable of handling both homophilic and heterophilic graphs. This is introduced in lines 67–68 and aligns with the terminology in classic works, e.g., [1]. In the revised manuscript, we will add clarifying footnotes at the first occurrence to eliminate any possible misunderstanding.
>
> Extending to link or graph-level tasks is valuable, yet these settings often introduce additional biases—such as subgraph sampling or graph-pooling choices—that can obscure the intrinsic topology–attribute fusion capability we study. We will leave broader evaluations to future work.
>
> [1] **Universal** Graph Convolutional Networks, NeurIPS 2021.
>
> - - -
>
> >W4&L3. While this design effectively decouples two fundamental sources of graph information, it is somewhat heuristic and lacks analysis on how this separation impacts performance across different graph types.
>
> R4. This design is mainly based on the theoretical insights from Theorems 3.2 and 3.3, which indicate that the potential mechanism of GTs is the cross-aggregation between topology and attributes. The implementation of cross-aggregation via cross-attention necessitates keeping topology and attribute representations separate before the cross-attention, i.e., at the beginning. Besides, this separating-then-combining scheme was validated in existing works, e.g., LINKX, which highlights the rationality of this design.
>
> - - -
>
> >W5&L4&Q2. UGCFormer is only evaluated with shallow architectures (1–5 layers), which is considerably lower than standard Transformer-based baselines (often using 6–12+ layers). Does your model mitigate over-smoothing?
>
> R5. Thanks for raising this important point. UGCFormer, like most node-level GTs (e.g., SGFormer and Polynormer), is evaluated with only a few layers (≤ 5) and exhibits a performance drop as depth increases (see the results below). Thus, UGCFormer does not benefit from deeper stacks.
>
>   - Cora: 84.9%→75.1%(16 L)→65.4%(32 L)→55.7% (64 L)
>   - CiteSeer: 73.4%→69.8%(16 L)→66.9%(32 L)→62.3% (64 L)
>
> Interestingly, graph-level GTs typically benefit from much deeper stacks (> 5 layers), aligning with the mentioned Transformers. We attribute this discrepancy to the **differing supervisory granularities of node-level task versus graph-level task**.
> - Node-level tasks: each node has its label. After shallow layers, the representation already encodes sufficient neighborhood context; deeper stacks merely average in distant, label-irrelevant signals.
> - Graph-level tasks: the entire graph carries a single label. Since each layer’s softmax averages all N nodes, the model needs to repeatedly refine and concentrate distant information into a single readout token.
>
>
> While we believe the performance drop in the above results is due to over-smoothing, GTs differ from local GNNs in practice. Local GNNs, which rely on message passing within local neighborhoods, often require stacking many layers to capture long-range dependencies, which can lead to over-smoothing. In contrast, GTs, with their inherent global expressiveness, can capture long-range dependencies **without needing many layers**.
>
> - - -
>
> >Q1. It's unclear whether UGCFormer can adapt to positional encodings or multi-hop connectivity representations.
>
> R6. The adjacency matrix is only the default input; the architecture can seamlessly integrate richer structural encodings. Specifically, replacing the adjacency matrix with (i) a positional encoding matrix (LapPE, SignNet, etc.) or (ii) a multi-hop propagation matrix such as the Personalized PageRank (PPR) matrix only requires a one-line change. We adopt the raw adjacency matrix primarily for simplicity. More sophisticated encodings inevitably incur heavy preprocessing: computing Laplacian eigenvectors (LapPE) costs O(N^3) in dense form and O(kN^2) even with sparse Lanczos, where N denotes the number of nodes and k represents a hyperparameter, while PPR requires O(N^3) for dense matrix powers or O(kE) under sparse-iterative acceleration. By relying on the unprocessed adjacency, we avoid these upfront costs while still delivering strong empirical results.
>
> - - -
>
> >Q3. I recommend that the authors carefully verify and test the provided code to ensure reproducibility and consistency with the descriptions in the paper.
>
> R7. Thanks for pointing out this error. In the released version, UGCFormer.forward() lacked the keyword argument 'adj_normal' to receive 'data.adj_normal' and assign it to the variable 'topology_' used in get_TopoRepresentation(). We sincerely apologize for the confusion caused by the incorrect release. Due to rebuttal rules, we cannot update the demo at this stage; we will release the fully corrected and reproducible GitHub repository as soon as the paper is accepted.
>
> >L5. There is no architectural analysis that clarifies why UGCFormer outperforms them. A discussion contrasting core design choices—such as attention mechanisms, position encoding, or expressivity—would improve the paper.
>
> R8. UGCFormer outperforms prior works by achieving a tighter alignment of topology and attributes. Existing models entangle a large number of intermediate representations (in Eq. 20), e.g., topology–attribute–attribute and attribute–topology–topology, producing an over-mixed, noisy embedding that weakens the final alignment and degrades performance. UGCFormer introduces the cross-attention-based DCA module to align the topology and attribute space and make every node selectively focus on the most relevant topology and attribute representations. Moreover, UGCFormer further enforces distributional alignment via a consistency loss. Together, they deliver superior performance.
>
>
>
> - - -
>
> >L6. An empirical analysis comparing efficiency with standard self-attention.
>
> R9. We have measured training time (ms) and GPU memory (MB) of DCA and two baselines with standard self-attention, i.e., Graphormer [1] and GraphTrans [2]. As shown in the result, DCA significantly reduces both training time and memory usage compared to standard self-attention, confirming its practical advantage.
>
> | | CiteSeer | PubMed | ogbn-arxiv |
> |:--------:|:---------:|:--------:| :--------:|
> | | Time / Mem. | Time / Mem. | Time / Mem. |
> | Graphormer | 478.0 / 870 | 870.4 / 1,844 | - / OOM |
> | GraphTrans | 85.6 / 528 | 118.2 / 1,460 | - / OOM |
> | UGCFormer | 8.8 /106 | 10.6 / 246 | 69.6 / 2,050 |
>
> [1] Do Transformers Really Perform Bad for Graph Representation?, NeurIPS 2021.
>
> [2] Representing Long-Range Context for Graph Neural Networks with Global Attention, NeurIPS 2021.
>
> - - -
>
> >L7. It would be helpful to analyze the component-wise impact on both homophilic and heterophilic graphs separately, or across varying graph sizes.
>
> R10. According to your suggestion, we have extended the ablation study to cover both homophilic and heterophilic graphs and varying graph sizes:
>
> | | UGCFormer | w/o CA | w/o L_con |
> |:--------:|:---------:|:--------:| :--------:|
> | Cora | 84.94 | 79.88 | 82.29 |
> | CiteSeer | 73.41 | 70.60 | 71.83 |
> | PubMed | 81.79 | 79.71 | 80.29 |
> | Cornell | 85.14 | 80.11 | 83.27 |
> | Texas | 84.59 | 81.29 | 83.71|
> | Wisconsin | 87.36 | 80.84 | 86.59 |
> | Actor | 37.41 | 36.77 | 37.09|
> | ogbn-proteins | 79.95 | 72.29 | 78.33|
> | ogbn-arxiv | 74.02 | 72.21 | 73.40 |
>
> Across all datasets, removing either component consistently hurts accuracy, confirming their necessity.
> - Cross-attention proves the most critical component: taking it away causes the steepest performance drops, in particular, the results on ogbn-proteins, and the heterophilic Cornell, Texas, and Wisconsin datasets, highlighting how well it learn distinctive node representations when the graph structure is noisy.
> - Consistency loss yields smaller but still clear gains, especially on smaller graphs (Cora, CiteSeer, PubMed), where it regularizes the dual-branch training; the gap shrinks on larger graphs (ogbn-arxiv, ogbn-proteins) yet remains positive.

---

> > ### Comment · Reviewer_3zmm · 2025-08-01
> >
> > Thank you for the detailed responses.

---

### Official Review · Reviewer_izVd · 2025-07-06

**Clarity:** 3
**Significance:** 2
**Originality:** 2
**Rating:** 4
**Confidence:** 3

**Summary:**

This paper reveals Cross Aggregation, a mechanism that effectively links graph topology and node attributes, by analyzing two main strategies utilizing by existing Graph Transformers. Based on this insight, UGCFormer is proposed, implementing the uncovered mechanism through a linearized Dual Cross-Attention module.

The key contributions of this paper include:

1.   **Identification of Cross-Aggregation Mechanism**. The paper explores the effectiveness of typical Graph Transformers and formalizes the cross-aggregation mechanism as a central component in GTs.
2.   **UGCFormer Architecture**. The paper proposes a linearized GT that utilizes the DCA module to aggregate graph topology and node attribute while maintaining linear time complexity.
3.   **Theoretical Foundations**. The paper provide proofs showing that Corss-Aggregation between topology and attributes is the essential mechanism in typical GTs.
4.   **Extensive Experiments**. The paper demonstrates the effectiveness of UGCFormer on multiple datasets, including both homophilic and heterophilic graphs, showing its effectiveness compared to existing models.

**Questions:**

1.   The paper explores its core mechanism by revisiting two main strategies used by existing Graph Transformers (GTs). Since this mechanism proves effective in this work, it suggests that a similar mechanism may also be playing a role in other GTs. However, the paper does not discuss the approaches of other related works. Could you provide specific methods from other related works and highlight how they differ from the approach in this paper?

2.   The paper introduces a novel Dual Cross-Attention module, but the exact dynamics of how the model balances information from the topology and attributes could be explained more intuitively. Could you provide a more detailed, intuitive explanation of how the DCA module updates representations and how it compares to other attention mechanisms used in graph learning (e.g., self-attention or cross-attention in NLP)?

3.   The paper briefly mentions that the consistency constraint helps mitigate overfitting. However, there is little discussion on how this mechanism performs under extreme cases, such as small datasets or highly noisy data. Have you conducted experiments where overfitting is more pronounced (e.g., using very small training sets or datasets with significant noise)?

4.   While the paper compares *UGCFormer* to multiple GNNs and GTs, the comparisons seem mostly quantitative. Can you provide a more detailed qualitative analysis, such as analyzing the impact of the cross-attention mechanism on the learned representations or providing case studies where *UGCFormer* performs better than its competitors?

**Ethical Concerns:**

["NO or VERY MINOR ethics concerns only"]

**Final Justification:**

The authors have addressed all my concerns.

**Limitations:**

Yes

**Paper Formatting Concerns:**

1.   line 338 in References Section, [12] has an extra year citation.
2.   line 228-229 in Section 3.3, the sentence which has syntax error is confusing.

**Quality:**

3

**Strengths And Weaknesses:**

## Strengths

1.   This paper provides a solid theoretical foundation by identifying and formalizing the concept of Cross-Aggregation between graph topology and node attributes. Such formal proofs strengthens the scientific rigor and ensures that the proposed method has a robust theoretical basis.
2.   The paper provides a thorough account of experimental setups, datasets, hyperparameters, and evaluation metrics, ensuring that others can replicate the experiments.
3.   *UGCFormer* introduces a practical solution that is likely to be valuable in various applications where scalability and efficiency are essential, by combining topology and attributes in a way that captures long-range dependencies without incurring high computational costs

## Weaknesses

1.   The core idea of combining graph topology and node attributes through attention module has been explored in other works, such as those involving GNNs with positional encodings or integrated GNN blocks. Therefore, while the specific design of UGCFormer is original, it is incremental innovation in attention mechanisms.
2.   The model propoesd requires tuning multiple hyperparameters, which may limit its ease of use in real-world applications. A more adaptive method for hyperparameter selection could improve the accessibility and usability of the model.
3.   While the experiments show impressive results on benchmark datasets, the paper could be improved by discussing potential applications in more depth. For example, how would *UGCFormer* perform on graph data with highly heterogeneous or noisy attributes? More concrete use cases or challenges faced when deploying the model in practical settings could be beneficial for demonstrating its significance.

---

> ### Author Rebuttal · Authors · 2025-07-31
>
> >W1. The core idea of combining graph topology and node attributes through an attention module has been explored in other works, such as those involving GNNs with positional encodings or integrated GNN blocks. Therefore, while the specific design of UGCFormer is original, it is an incremental innovation in attention mechanisms.
>
> R1. This may be a misunderstanding of our contribution. As we know, effectively integrating graph topology and node attributes has always been a key topic in graph learning models. However, they always do outside the attention module: Graph Neural Networks (GNNs) propagate attributes over the graph topology, and Graph Transformers concatenate positional encodings (topology representations) to node attributes. In contrast, this is the first work to explore and reveal that the essence of these two methods is to capture the interrelation between graph topology and node attributes through a common mechanism, and to design a simple yet effective cross-attention module accordingly.
>
> Moreover, unlike the attention module in existing GTs, which focuses on correlations within a single representation space, our proposed attention performs fusion between topology and attribute spaces. This represents a significant contribution that clearly distinguishes our attention mechanism from existing ones. We hope it addresses your concerns regarding design novelty.
>
> - - -
>
> >W2. The model proposed requires tuning multiple hyperparameters, which may limit its ease of use in real-world applications. A more adaptive method for hyperparameter selection could improve the accessibility and usability of the model.
>
> R2. Thank you for highlighting the importance of training efficiency—this is indeed a shared limitation of current GTs. We have already noted it in Section F and regard it as a priority for future work.
>
> - - -
>
>
> >W3. How would UGCFormer perform on graph data with highly heterogeneous or noisy attributes? More concrete use cases or challenges faced when deploying the model in practical settings could be beneficial for demonstrating its significance.
>
> R3. Thank you for your valuable feedback, which has guided us to reflect on the model's robustness and practical deployment. To verify the robustness of UGCFormer, we have conducted robustness experiments on two datasets (Cora and CiteSeer). We adopt two simple noise settings:
> (1) Randomly masking $p\%$ of the attribute dimensions across all nodes;
> (2) Injecting Gaussian noise ($\sigma = 0.1$) into the attributes of $p\%$ of randomly chosen nodes. The results are as follows.
>
> | p | 0 | 20 | 40 | 60 | 80 |
> |:--------|:---------:|:--------:| :--------:| :--------:| :--------:|
> **Masking**
> | Cora | 84.9 | 81.3 | 78.3 | 73.7 | 67.1 |
> | CiteSeer | 73.4 | 65.5 | 62.6 | 59.7 | 55.9|
> **Injecting Noise**
> | Cora | 84.9 | 81.7 | 79.9 | 76.9 | 79.5 |
> | CiteSeer | 73.4 | 70.1 | 68.4 | 65.3 | 67.0 |
>
> Across all cases, UGCFormer maintains stable performance, demonstrating its robustness to attribute noise. This stems from the dual-branch design between the topology and attribute spaces, which treats topology as a stabilizing prior: when attributes are noisy, cross-attention and the consistency loss use the topology to suppress attribute noise and guide the learning of attribute representations.
>
> For highly heterogeneous attributes, UGCFormer can ingest multi-modal inputs through modality-specific encoders followed by a shared projection layer, allowing the cross-attention to fuse diverse signals without manual alignment. We plan to pursue this as future work.
>
> - - -
>
> >Q1. The paper explores its core mechanism by revisiting two main strategies used by existing Graph Transformers (GTs). Since this mechanism proves effective in this work, it suggests that a similar mechanism may also be playing a role in other GTs. However, the paper does not discuss the approaches of other related works. Could you provide specific methods from other related works and highlight how they differ from the approach in this paper?
>
> R4. According to your suggestion, we have explored the relationship between the two other categories of GTs, namely, those involving sampling context nodes and those involving rewriting edges, and the cross-aggregation mechanism, as follows.
>
> Although both categories of GTs are implemented in different ways, their essence is to determine the subgraph for each node to perform local message passing (via graph convolution or self-attention). Thus, they can be uniformly expressed by obtaining the graph structure. As a result, the topology representation can be straightforwardly determined based on the eigenvalue decomposition of the adjacency matrix, as described in the manuscript. Ultimately, a formulation similar to that of cross-aggregation can be derived. Thus, their underlying mechanism can be attributed to a cross-aggregation between topology and attribute representations.
> Thank you for guiding us to further discuss a broader category of GTs. We appreciate the opportunity to further enhance the comprehensiveness and generalizability of our theoretical contributions. We will add the above discussion to the revised manuscript.
> - - -
>
> >Q2. Could you provide a more detailed, intuitive explanation of how the DCA module updates representations and how it compares to other attention mechanisms used in graph learning (e.g., self-attention or cross-attention in NLP)?
>
> R5. The DCA module intuitively balances information from topology and attributes through an optimization framework (in Theorem 3.4) that explicitly models their interrelations. Firstly, the third term, which uses the Hilbert-Schmidt Independence Criterion (HSIC), measures the dependence between topology representations (from the first term) and attribute representations (from the second term). This allows the model to learn whether to correlate or exclude certain aspects of topology and attributes, thus modulating their interrelation. Secondly, the balance between topology and attributes can be achieved by tuning the parameters $λ$ and $η$, which control the trade-off between these three terms. By minimizing the objective function, the DCA module learns to integrate topology and attributes in a way that best captures the underlying structure and semantics of the graph.
>
> The DCA module is designed to update representations by capturing the interrelations between graph topology and node attributes. This dual attention process allows for a more nuanced integration of topology and attributes compared to traditional self-attention in graph learning, which is typically without explicit attribute-topology alignment. Moreover, unlike cross-attention in NLP, which handles sequences of tokens, DCA is tailored for graph-structured data, addressing the unique challenge of integrating topology and attributes. Thus, DCA enables learning more expressive representations suited for diverse graph learning tasks.
>
> - - -
>
> >Q3. There is little discussion on how this mechanism performs under extreme cases, such as small datasets or highly noisy data. Have you conducted experiments where overfitting is more pronounced (e.g., using very small training sets or datasets with significant noise)?
>
> R6. According to your suggestion, we have conducted an overfitting study to investigate the effect of the consistency constraint. We compared UGCFormer with a variant without the proposed consistency constraint, using only 1, 3, 5, 10, and 15 labeled nodes per class. We observe that the variant without the consistency term suffers a steep accuracy decline as fewer labels are provided, whereas UGCFormer’s performance degrades modestly. This confirms that the proposed constraint effectively curbs overfitting under extreme label scarcity. This gain is mainly due to the consistency loss, which causes the topology and attribute representations to supervise each other. Full results are reported in the following Table.
>
> | | 1 | 3 | 5 | 10 | 15 |
> |:--------|:---------:|:--------:| :--------:| :--------:| :--------:|
> **Cora**
> | w/o L_con | 41.7 | 62.4 | 70.1 |  75.4 | 79.9  |
> | UGCFormer | 50.9 | 70.5 | 76.4 | 78.7 | 81.2 |
> **CiteSeer**
> | w/o L_con | 36.8 | 48.2 |  59.1 | 66.3 | 69.8 |
> | UGCFormer | 45.1 | 53.7 | 65.2 |  69.0 | 72.2 |
>
> - - -
>
> >Q4. Can you provide a more detailed qualitative analysis, such as analyzing the impact of the cross-attention mechanism on the learned representations or providing case studies where UGCFormer performs better than its competitors?
>
> R7. To illustrate the impact of the cross-attention mechanism on learned representations, we have added the following qualitative study. Specifically, we use t-SNE to visualize the node embeddings generated by UGCFormer and by LINKX, which can be viewed as a UGCFormer's variant without cross-attention.
>
> Compared with LINKX, UGCFormer yields tighter intra-class clusters and larger inter-class margins, indicating that the cross-attention mechanism successfully disentangles topology and attribute and produces more discriminative representations.
>
> Due to constraints, we are unable to include external links or images here; the complete t-SNE visualizations will be provided in the revised manuscript.
>
> - - -
>
> > Paper Formatting Concerns: the extra year citation in the References [12] and the syntax error in Lines 228-229.
>
> R8. Thank you for your careful review. We have corrected the identified issues and conducted a thorough review of the entire manuscript.

---

> > ### Comment · Reviewer_izVd · 2025-08-05
> >
> > Thanks for the detailed response.

---

### Comment · Area_Chair_tCNJ · 2025-08-07

Dear Reviewer,

Thank you very much for your diligent work in reviewing submissions for NeurIPS. We are now approaching the end of the Author-Reviewer Discussion period. Please be reminded that according to the NeurIPS guidelines this year, reviewers must actively participate in discussions with authors before submitting the "Mandatory Acknowledgement." **Simply submitting the "Mandatory Acknowledgement" without sufficient engagement in these discussions is not acceptable**.

If you have not yet responded to the authors, please do so as soon as possible, ensuring sufficient time for meaningful discussions.

Please note that insufficient participation may impact your own submissions under this year's Responsible Reviewing Initiative.

Thank you again for your valuable contributions!


AC

---

### Decision · Program_Chairs · 2025-09-17

**Decision:**

Accept (spotlight)

**Comment:**

**Summary:** This paper proposes UGCFormer, a universal graph Transformer that formalizes the cross-aggregation mechanism as the key driver of existing Graph Transformers and implements it with a linearized Dual Cross-Attention (DCA) module. The model explicitly integrates graph topology and node attributes, with a consistency constraint to alleviate overfitting. The paper provides theoretical justification for the mechanism, comprehensive experiments across homophilic and heterophilic benchmarks, ablation studies, and efficiency analyses.

**Decision:** Reviewers found the work technically solid and the proposed architecture both effective and efficient. Some concerns were raised regarding novelty, the need for clearer distinction from related works, code reproducibility, and analysis of underperformance on certain datasets. The authors engaged actively during rebuttal, offering additional robustness experiments, qualitative analyses, and clarifications on theoretical and implementation aspects. These responses resolved most concerns, and reviewers updated their ratings accordingly. Overall, the paper makes a clear and valuable contribution to the development of graph Transformers. I therefore recommend acceptance.